# The Missing U for Efficient Diffusion Models

**Sergio Calvo-Ordoñez**[1,2], **Chun-Wun Cheng**[3], **Jiahao Huang**[4], **Lipei Zhang**[3], **Guang Yang**[4,5,6],
**Carola-Bibiane Schönlieb**[3], **Angelica I Aviles-Rivero**[3]

[1] *Oxford-Man Institute of Quantitative Finance, University of Oxford*

[2] *Mathematical Institute, University of Oxford*

[3] *Department of Applied Mathematics and Theoretical Physics, University of Cambridge*

[4] *Bioengineering Department and Imperial-X & National Heart and Lung Institute, Imperial College London*

[5] *Cardiovascular Research Centre, Royal Brompton Hospital London*

[6] *School of Biomedical Engineering Imaging Sciences, King's College London*

**Reviewed on OpenReview:** https://openreview.net/forum?id=Y4YWzBiTEV

## Abstract

Diffusion Probabilistic Models stand as a critical tool in generative modelling, enabling the generation of complex data distributions. This family of generative models yields record-breaking performance in tasks such as image synthesis, video generation, and molecule design. Despite their capabilities, their efficiency, especially in the reverse process, remains a challenge due to slow convergence rates and high computational costs. In this paper, we introduce an approach that leverages continuous dynamical systems to design a novel denoising network for diffusion models that is more parameter-efficient, exhibits faster convergence, and demonstrates increased noise robustness. Experimenting with Denoising Diffusion Probabilistic Models (DDPMs), our framework operates with approximately a quarter of the parameters, and $\sim 30\%$ of the Floating Point Operations (FLOPs) compared to standard U-Nets in DDPMs. Furthermore, our model is notably faster in inference than the baseline when measured in fair and equal conditions. We also provide a mathematical intuition as to why our proposed reverse process is faster as well as a mathematical discussion of the empirical tradeoffs in the denoising downstream task. Finally, we argue that our method is compatible with existing performance enhancement techniques, enabling further improvements in efficiency, quality, and speed.

## 1 Introduction

Diffusion Probabilistic Models, grounded in the work of (Sohl-Dickstein et al., 2015) and expanded upon by (Song & Ermon 2020; Ho et al. 2020; Song et al. 2020b), have achieved remarkable results in various domains, including image generation (Dhariwal & Nichol, 2021; Nichol & Dhariwal, 2021; Ramesh et al., 2022; Saharia et al., 2022; Rombach et al., 2022b), audio synthesis (Kong et al., 2021; Liu et al., 2022a), and video generation (Ho et al., 2022; Ho et al., 2021). These score-based generative models utilise an iterative sampling mechanism to progressively denoise random initial vectors, offering a controllable trade-off between computational cost and sample quality. Although this iterative process offers a method to balance quality with computational expense, it often leans towards the latter for state-of-the-art results. Generating top-tier samples often demands a significant number of iterations, with the diffusion models requiring up to 2000 times more computational power compared to other generative models (Goodfellow et al., 2020; Kingma & Welling, 2013; Rezende et al., 2014; Rezende & Mohamed, 2015; Kingma & Dhariwal, 2018).

Recent research has delved into strategies to enhance the efficiency and speed of this reverse process. In Early-stopped Denoising Diffusion Probabilistic Models (ES-DDPMs) proposed by (Lyu et al., 2022), the diffusion process is stopped early. Instead of diffusing the data distribution into a Gaussian distribution via hundreds of iterative steps, ES-DDPM considers only the initial few diffusion steps so that the reverse denoising process

starts from a non-Gaussian distribution. A similar method used in (Zheng et al., 2023b), also truncates the forward process allowing for fewer reverse steps to generate the data. Additionally (Xiao et al., 2022) reduce the sampling process overhead by modelling the denoising distribution using a complex multimodal distribution with a denoising diffusion generative adversarial network for each step. (Lu et al., 2022) propose an exact formulation of the solution of diffusion ODEs, allowing sampling in a few steps. Continuing with faster sampling methodologies, (Zhang & Chen, 2023) present *Diffusion Exponential Integrator Sampler* which leverages a semilinear structure of the learned diffusion process to reduce the discretisation error and is more efficient. Another significant contribution is the Analytic-DPM framework (Bao et al., 2022). This training-free inference framework estimates the analytic forms of variance and Kullback-Leibler divergence using Monte Carlo methods in conjunction with a pre-trained score-based model. Results show improved log-likelihood and a speed-up between 20x to 80x. Furthermore, approaches that study using manifold constraints and inverse problems hypothesis for diffusion models (Chung et al. 2022; Liu et al 2022b; Chung et al. 2023; Rout et al. 2023; Lou & Ermon 2023) achieve a significant performance boost. Other lines of work focused on modifying the sampling process during the inference while keeping the model unchanged. (Song et al., 2020a) proposed Denoising Diffusion Implicit Models (DDIMs) where the reverse Markov chain is altered to take deterministic *jumping* steps composed of multiple standard steps. This reduces the steps required but may introduce discrepancies from the original diffusion process. (Nichol & Dhariwal, 2021) proposed timestep respacing to non-uniformly select timesteps in the reverse process. While reducing the total number of steps, can cause deviation from the model's training distribution. In general, these methods provide inference-time improvements but do not accelerate model training.

A different approach trains diffusion models with continuous timesteps and noise levels to enable variable numbers of reverse steps after training (Song & Ermon, 2020). Models trained directly on continuous objectives outperform discretely-trained models on continuous data where the score function is properly defined (Song et al. 2020b; Karras et al. 2022). (Kong et al., 2021) approximate continuous noise levels through interpolation of discrete timesteps, but lack theoretical grounding. Orthogonal strategies accelerate diffusion models by incorporating conditional information. (Preechakul et al., 2022a) inject an encoder vector to guide the reverse process. While effective for conditional tasks, it provides limited improvements for unconditional generation. (Salimans & Ho, 2022) distil a teacher model into students taking successively fewer steps, reducing steps without retraining, but distillation cost scales with teacher steps. Unlike existing work, we underline that our approach diverges by parameterising the dynamics via a second-order ODE that specifically models acceleration in the reverse process.

To tackle these issues, throughout this paper, we construct and evaluate an approach that rethinks the reverse process in diffusion models by fundamentally altering the denoising network architecture. Current literature predominantly employs U-Net architectures for the discrete denoising of diffused inputs over a specified number of steps. Many reverse process limitations stem directly from constraints inherent to the chosen denoising network. Building on the work of (Cheng et al., 2023), we leverage continuous dynamical systems to design a novel denoising network that is parameter-efficient, exhibits faster and better convergence, demonstrates robustness against noise, and outperforms conventional U-Nets while providing theoretical underpinnings. We show that our architectural shift directly enhances the reverse process of diffusion models by offering comparable performance in image synthesis but an improvement in inference time in the reverse process, denoising performance, and operational efficiency. Importantly, our method is orthogonal to existing performance enhancement techniques, allowing their integration for further improvements. Furthermore, we delve into a mathematical discussion to provide a foundational intuition as to why it is a sensible design choice to use our deep implicit layers in a denoising network that is used iteratively in the reverse process. Along the same lines, we empirically investigate our network's performance at sequential denoising and theoretically justify the tradeoffs observed in the results. Besides our framework's compatibility with other families of diffusion models (as discussed in Section 3), the method could be leveraged for downstream tasks in other areas, for example, and not limited to, MRI reconstruction, audio generation, image segmentation, or synthetic data generation. In particular, our contributions are:

- We propose a new denoising network that incorporates an original dynamic Neural ODE block integrating residual connections and time embeddings for the temporal adaptivity required by diffusion models.

- We develop a novel family of diffusion models that uses a deep implicit U-Net denoising network; as an alternative to the standard discrete U-Net and achieve enhanced efficiency.

- We evaluate our framework, demonstrating competitive performance in image synthesis, and perceptually outperforms the baseline in denoising with approximately 4x fewer parameters, smaller memory footprint, and shorter inference times.

## 2 Preliminaries

This section provides a summary of the theoretical ideas of our approach, combining the strengths of continuous dynamical systems, continuous U-Net architectures, and diffusion models.

**Denoising Diffusion Probabilistic Models (DDPMs).** These models extend the framework of DPMs through the inclusion of a denoising mechanism (Ho et al., 2020). The latter is used an inverse mechanism to reconstruct data from a latent noise space achieved through a stochastic process (reverse diffusion). This relationship emerges from (Song et al., 2020b), which shows that a certain parameterization of diffusion models reveals an equivalence with denoising score matching over multiple noise levels during training and with annealed Langevin dynamics during sampling. DDPMs can be thought of as analog models to hierarchichal VAEs (Cheng et al., 2020), with the main difference being that all latent states, $x_t$ for $t = [1, T]$, have the same dimensionality as the input $x_0$. This detail makes them also similar to normalizing flows (Rezende & Mohamed, 2015), however, diffusion models have hidden layers that are stochastic and do not need to use invertible transformations.

**Neural ODEs.** Neural Differential Equations (NDEs) offer a continuous-time approach to data modelling (Chen et al., 2018). They are unique in their ability to model complex systems over time while efficiently handling memory and computation (Rubanova et al., 2019). A Neural Ordinary Differential Equation is a specific NDE described as:

$$y(0) = y_0, \qquad \frac{dy}{dt}(t) = f_\theta(t, y(t)), \tag{1}$$

where $y_0 \in \mathbb{R}^{d_1 \times \cdots \times d_k}$ refers to an input tensor with any dimensions, $\theta$ symbolizes a learned parameter vector, and $f_\theta : \mathbb{R} \times \mathbb{R}^{d_1 \times \cdots \times d_k} \to \mathbb{R}^{d_1 \times \cdots \times d_k}$ is a neural network function. Typically, $f_\theta$ is parameterized by simple neural architectures, including feedforward or convolutional networks. The selection of the architecture depends on the nature of the data and is subject to efficient training methods, such as the adjoint sensitivity method for backpropagation through the ODE solver.

**Continuous U-Net.** (Cheng et al., 2023) propose a new U-shaped network for medical image segmentation motivated by works in deep implicit learning and continuous approaches based on neural ODEs (Chen et al., 2018; Dupont et al., 2019). This novel architecture consists of a continuous deep network whose dynamics are modelled by second-order ordinary differential equations. The idea is to transform the dynamics in the network - previously CNN blocks - into dynamic blocks to get a solution. This continuity comes with strong and mathematically grounded benefits. Firstly, by modelling the dynamics in a higher dimension, there is more flexibility in learning the trajectories. Therefore, continuous U-Net requires fewer iterations for the solution, which is more computationally efficient and in particular provides constant memory cost. Secondly, it can be shown that continuous U-Net is more robust than other variants (CNNs), and (Cheng et al., 2023) provides an intuition for this. Lastly, because continuous U-Net is always bounded by some range, unlike CNNs, the network is better at handling the inherent noise in the data.

## 3 Related Work

A significant amount of research focuses on speeding up diffusion models. In this section, we offer an overview of these strategies and detail how our method distinctly contributes to improving diffusion model efficiency. We also argue our method's orthogonality by outlining potential ways to incorporate our framework into existing works, underscoring its compatibility and additive impact on the field.

The work introduced by (Wang et al., 2022), introduces an efficient diffusion model tailored for generating diverse textures and patterns from a single image, leveraging a unique approach to learn the distribution of image patches, enabling high-quality synthesis with minimal data. Patch Diffusion (Wang et al., 2023), introduces a novel training approach for diffusion models, significantly reducing training time and enhancing data efficiency by learning conditional score functions on image patches of varying sizes and locations. (Zheng et al., 2023a) look into accelerating diffusion models through a novel sampling technique that uses neural operators to solve the probability flow ODE. (Arakawa et al., 2023) present a proposal that uses a patch-based diffusion probabilistic model that divides the images into patches and generates them independently to reduce memory consumption during inference. Each of these methods introduces unique efficiencies within the diffusion modelling framework, yet they all use U-Net architectures for the denoising process, therefore providing benefits which are independent of the denoising network choice. This suggests that there is room for improvement through the implementation of our method to further increase efficiency and reduce memory requirements while preserving image quality.

(Gao et al., 2023) and (Zheng et al., 2023c) present novel approaches to enhance diffusion models, with the former introducing a mask latent modelling scheme for improved learning speed and the latter focusing on integrating local features and global content for efficiency gains. Despite their advancements, (Gao et al., 2023) report no increase in memory efficiency, and the methodology in (Zheng et al., 2023c) has not been extensively tested across all diffusion model applications, in some cases resulting in lower quality outputs.

Furthermore, there is a branch of literature that explores applying the efficiency benefits of variational autoencoders (VAEs) to diffusion models. (Preechakul et al., 2022b) presents a method combining a learnable encoder for capturing high-level semantics with a diffusion probabilistic model (DPM) as the decoder for modelling stochastic variations, showing improved efficiency and generation quality over DDIMs. However, its performance relative to newer methods and its compatibility with other acceleration techniques remains unclear, as the diffusion process is secondary to encoding. (Pandey et al., 2022) presents a framework that integrates the ability to operate in a low-dimensional latent space through VAEs while still modelling a stochastic process with diffusion mechanisms. Similarly, (Rombach et al., 2022a), apply diffusion models in the latent space of powerful pre-trained autoencoders. However, this is limited by the bottleneck of the pre-trained autoencoder. In all of these cases, their use of a U-Net architecture for denoising aligns with our framework, suggesting the potential for integration to enhance efficiency further. (Vahdat et al., 2021) train Score-based Generative Models (SGM) in latent space to make them more efficient with the reduction of the dimensionality. However, although improving upon the inefficiencies of diffusion processes, it still is considerably slower than similar techniques applied in discrete-time frameworks.

Lastly, other works introduce general methods to improve different stages of diffusion models. Karras et al. (2022) propose an analysis that helps simplify the stages of diffusion models that use neural networks (e.g. U-Nets) to model the score of a noise level-dependent marginal distribution of the training data corrupted by noise. Dockhorn et al. (2022) introduce critically-damped Langevin diffusion simplifying the score-matching process by only needing to learn the score-function of the conditional distribution of a subset of the variables. Finally, (Pandey & Mandt, 2023) introduce a similar concept, Phase Space Langevin Diffusion (PSLD). This novel SGM enhances sample quality and optimises the speed-quality trade-off by performing diffusion in an augmented space with auxiliary variables. However, this technique is only applicable in frameworks that use fully continuous points of view of the forward and reverse processes.

Below, we describe our methodology and where each of the previous concepts plays an important role within our proposed model architecture.

## 4  Methodology

In standard DDPMs, the reverse process involves reconstructing the original data from noisy observations through a series of discrete steps using variants of a U-Net architecture. In contrast, our approach (Fig. 1) employs a continuous U-Net architecture to model the reverse process in a *locally continuous-time setting*[1].

---

[1]The *locally continuous-time setting* denotes a hybrid method where the main training uses a discretised framework, but each step involves continuous-time modelling of the image's latent representation, driven by a neural ordinary differential equation.

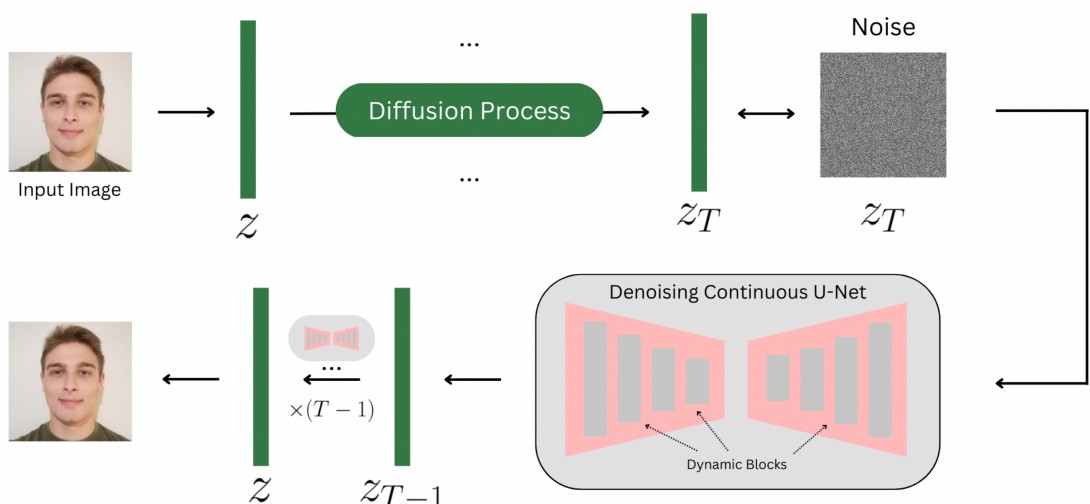

Figure 1: Visual representation of our framework featuring implicit deep layers tailored for denoising in the reverse process of a DDPM, enabling the reconstruction of the original data from a noise-corrupted version.

Unlike previous work on continuous U-Nets, focusing on segmentation (Cheng et al., 2023), we adapt the architecture to carry out denoising within the reverse process of DDPMs, marking the introduction of the first continuous U-Net-based denoising network. Our changes touch upon:

● **Model Adaptation for Denoising:** We have reconfigured the architecture to better suit denoising tasks. This includes adjustments in output channels, transition from a categorical cross-entropy loss to a reconstruction-based loss for minimising pixel discrepancies, and stride modifications to preserve spatial resolution.

● **Temporal Dynamics Integration:** Time embeddings have been introduced, following the approach by (Ho et al., 2020), to accurately model and adapt to the diffusion process across time, enhancing the model's ability to dynamically adjust to various diffusion stages.

● **Architectural Innovations:** Our continuous U-Net now incorporates attention mechanisms and residual connections, aiming to capture long-range dependencies and improve noise management capabilities, marking a departure from traditional designs towards a more dynamic and efficient architecture.

Overall, our architecture is strategically rooted in its capability to significantly reduce computational cost without increasing it. This reduction is achieved through the decreased necessity for storing active functions and leveraging the adjoint sensitivity method, which guarantees $\mathcal{O}(1)$ memory cost regardless of model complexity. This approach inherently builds reversibility into the architecture, ensuring efficient memory usage and substantially reducing computation time.

### 4.1 Dynamic Blocks for Diffusion

Our dynamical blocks are based on second-order ODEs, therefore, we make use of an initial velocity block that determines the initial conditions for our model. We leverage instance normalisation, and include sequential convolution operations to process the input data and capture detailed spatial features. The first convolution transitions the input data into an intermediate representation, then, further convolutions refine and expand the feature channels, ensuring a comprehensive representation of the input. In between these operations, we include ReLU activation layers to enable the modelling of non-linear relationships as a standard practice due to its performance (Agarap, 2019).

Furthermore, our design incorporates a neural network function approximator block (Fig. 2 - right), representing the derivative in the ODE form $\frac{dz}{dt} = f(t, z)$ which dictates how the hidden state $z$ evolves over

the continuous-time variable $t$. Group normalisation layers are employed for feature scaling, followed by convolutional operations for spatial feature extraction. In order to adapt to diffusion models, we integrate time embeddings using multi-layer perceptrons that adjust the convolutional outputs via scaling and shifting and are complemented by our custom residual connections. Additionally, we use an ODE block (Fig. 2 - left) that captures continuous-time dynamics, wherein the evolutionary path of the data is defined by an ODE function and initial conditions derived from preceding blocks.

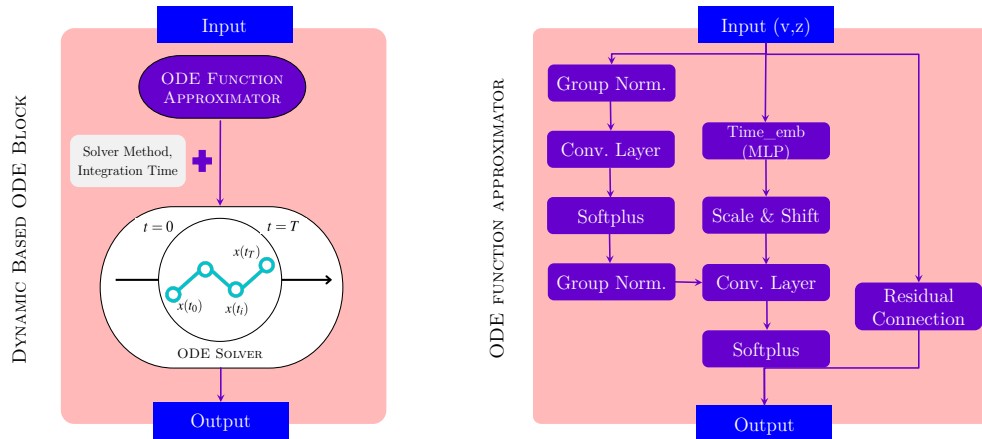

Figure 2: Architectural components of the continuous U-Net. On the left, the Dynamic ODE Block represents the core unit of our continuous model, detailing the integration of the ODE solver and function approximator within the network structure. The right panel expands on the ODE Function Approximator, highlighting the convolutional layers, group normalisation, time embeddings, and the incorporation of scale, shift operations, and residual connections to accurately adapt the network's dynamics during the diffusion process.

## 4.2 A New 'U' for Diffusion Models

As we fundamentally modify the denoising network used in the reverse process, it is relevant to look into how the mathematical formulation of the reverse process of DDPMs changes. The goal is to approximate the transition probability using our model. Denote the output of our continuous U-Net as $\tilde{U}(x_t, t, \tilde{t}; \Psi)$, where $x_t$ is the input, $t$ is the time variable related to the DDPMs, $\tilde{t}$ is the time variable related to neural ODEs and $\Psi$ represents the parameters of the network including $\theta_f$ from the dynamic blocks built into the architecture. We use the new continuous U-Net while keeping the same sampling process (Ho et al., 2020) which reads

$$x_{t-1} = \frac{1}{\sqrt{\alpha_t}} \left( x_t - \sqrt{\beta_t} \frac{1}{\sqrt{1 - \bar{\alpha}_t}} \epsilon_\theta(x_t, t) \right) + \sigma_t z, \text{ where } z \sim \mathcal{N}(0, I) \tag{2}$$

As opposed to traditional discrete U-Net models, this reformulation enables modelling the transition probability using the continuous-time dynamics encapsulated in our architecture. Going further, we can represent the continuous U-Net function in terms of dynamical blocks given by:

$$\epsilon_\theta(x_t, t) \approx \tilde{U}(x_t, t, \tilde{t}; \theta) \tag{3}$$

where,

$$\begin{cases} x''_{\tilde{t}} = f^{(a)}(x_{\tilde{t}}, x'_{\tilde{t}}, t, \tilde{t}, \theta_f) \\ x_{\tilde{t}_0} = X_0, \quad x'_{\tilde{t}_0} = g(x_{\tilde{t}_0}, \theta_g) \end{cases} \tag{4}$$

Here, $x''_t$ represents the second-order derivative of the state with respect to time (acceleration), $f^{(a)}(\cdot, \cdot, \cdot, \theta_f)$ is the neural network parametrising the acceleration and dynamics of the system, and $x_{t_0}$ and $x'_{t_0}$ are the

initial state and velocity. $X_0$ is the initial value and $g(x_{\tilde{t}_0}, \theta_g)$ is the neural network parameterising the velocity. Then we can update the iteration by $x_t$ to $x_{t-1}$ by the continuous network.

## 4.3 Unboxing the Missing U for Faster and Lighter Diffusion Models

Our architecture outperformed DDPMs in terms of efficiency and accuracy (Table 1). This section provides a mathematical justification for the performance. We first show that the Probability Flow ODE is faster than the stochastic differential equation (SDE). This is shown when considering that the SDE can be viewed as the sum of the Probability Flow ODE and the Langevin Differential SDE in the reverse process (Karras et al., 2022). We can then define the continuous reverse SDE (Song et al., 2020b) as:

$$dx_t = [f(x_t, t) - g(t)^2 \nabla_{x_t} \log p_t(x_t)]dt + g(t)dw_t \tag{5}$$

We can also define the probability flow ODE as follows:

$$dx_t = [f(x_t, t) - g(t)^2 \nabla_{x_t} \log p_t(x_t)]dt \tag{6}$$

We can reformulate the expression by setting $f(x_t, t) = -\frac{1}{2}\beta(t)x_t$, $g(t) = \sqrt{\beta(t)}$ and $s_{\theta_b}(x_t) = \nabla_x \log p_t(x_t)$. Substituting these into (5) and (6) yields the following two equations for the SDE and Probability Flow ODE, respectively.

$$dx_t = -\frac{1}{2}\beta(t)[x_t + 2s_{\theta_b}(x_t)]dt + \sqrt{\beta(t)}dw_t \tag{7}$$

$$dx_t = -\frac{1}{2}\beta(t)[x_t + s_{\theta_b}(x_t, t)]dt \tag{8}$$

We can then perform the following operation:

$$\begin{aligned}
dx_t &= -\frac{1}{2}\beta(t)[x_t + 2s_{\theta_b}(x_t)]dt + \sqrt{\beta(t)}dw_t \\
&= -\frac{1}{2}\beta(t)[x_t + s_{\theta_b}(x_t)]dt - \frac{1}{2}\beta(t)s_{\theta_b}(x_t, t)dt + \sqrt{\beta(t)}dw_t
\end{aligned} \tag{9}$$

Expression (9) decomposes the SDE into the Probability Flow ODE and the Langevin Differential SDE. This indicates that the Probability Flow ODE is faster, as discretising the Langevin Differential equation is time-consuming. However, we deduce from this fact that although the Probability Flow ODE is faster, it is less accurate than the SDE. This is a key reason for our interest in second-order neural ODEs, which can enhance both speed and accuracy. Notably, the Probability Flow ODE is a form of first-order neural ODEs, utilising an adjoint state during backpropagation. But what exactly is the adjoint method in the context of Probability Flow ODE? To answer this, we give the following proposition.

> **Proposition**
>
> **Proposition 4.1** *The adjoint state $r_t$ of probability flow ODE follows the first order ODE*
>
> $$r'_t = -r_t^T \frac{\partial \frac{1}{2}\beta(t)[-x_t - s_{\theta_b}(x_t, t)]}{\partial X_t} \tag{10}$$

*Proof.* Following (Norcliffe et al., 2020), we denote the scalar loss function be $L = L(x_{t_n})$, and the gradient respect to a parameter $\theta$ as $\frac{dL}{d\theta} = \frac{\partial L}{\partial x_{t_n}} \cdot \frac{dx_{t_n}}{d\theta}$. Then $x_{t_n}$ follows:

$$\begin{cases} x_{t_n} = \int_{t_0}^{t_n} x'_t dt + x_{t_0} \\ x_{t_0} = f(X_0, \theta_f), \quad x'_t = \frac{1}{2}\beta(t)[-x_t - s_{\theta_b}(x_t, t)] \end{cases} \tag{11}$$

Let $\boldsymbol{K}$ be a new variable such that satisfying the following integral:

$$\boldsymbol{K} = \int_{t_0}^{t_n} x'_t dt$$

$$= \int_{t_0}^{t_n} \left( x'_t + A(t)[x'_t - \frac{1}{2}\beta(t)[-x_t - s_{\theta_b}(x_t, t)]] \right) dt + B(x_{t_0} - f) \tag{12}$$

Then we can take derivative of $\boldsymbol{K}$ respect to $\theta$

$$\frac{d\boldsymbol{K}}{d\theta} = \int_{t_0}^{t_n} \frac{x'_t}{d\theta} dt + \int_{t_0}^{t_n} A(t) \left( \frac{dx'_t}{d\theta} - \frac{\partial[\frac{1}{2}\beta(t)[-x_t - s_{\theta_b}(x_t, t)]}{\partial\theta} - \frac{\partial[\frac{1}{2}\beta(t)[-x_t - s_{\theta_b}(x_t, t)]}{\partial x^T} \right) dt$$
$$+ B\left( \frac{dx_{t_0}}{d\theta} - \frac{df}{d\theta} \right) \tag{13}$$

Use the freedom of choice of A(t) and B, then we can get the following first-order adjoint state.

$$r'_t = -r_t^T \frac{\partial \frac{1}{2}\beta(t)[-x_t - s_{\theta_b}(x_t, t)]}{\partial X_t} \tag{14}$$

∎

As observed, the adjoint state of the Probability Flow ODE adheres to the first-order ODE, where gradients are calculated by performing backward integration of both the adjoint state, $r$, and the real state, $x$, through time. This approach not only obviates the need to store intermediate values—thereby utilising a fixed amount of memory and offering a substantial advantage over traditional backpropagation methods—but also lays the groundwork for our model's efficiency. By repurposing the first-order adjoint method within our second-order neural ODE framework, we significantly enhance computational efficiency. This strategic choice is rooted in the finding that the computation cost of the second-order adjoint method is, at a minimum, comparable to that of the first-order adjoint method, with the latter often requiring less computation time and cost. Such an integration of the probability flow ODE with our model architecture directly contributes to improved accuracy and speed, leveraging the universal approximation theorem, higher differentiability, and the expanded flexibility of second-order neural ODEs for transformations beyond homeomorphic shifts in real space.

There is still a final question in mind, the probability flow ODE is for the whole model but our continuous U-Net optimises in every step. What is the relationship between our approach and the DDPMs? This can be answered by a concept from numerical methods. If a given numerical method has a local error of $O(h^{k+1})$, then the global error is $O(h^k)$. This indicates that the order of local and global errors differs by only one degree. To better understand the local behaviour of our DDPMs, we aim to optimise them at each step. This approach, facilitated by a continuous U-Net, allows for a more detailed comparison of the order of convergence between local and global errors.

## 5  Experimental Results

In this section, we detail the set of experiments to validate our proposed framework.

### 5.1  Image Synthesis

We evaluated our method's efficacy via generated sample quality (Fig. 3). As a baseline, we used a DDPM that uses the same U-Net described in (Ho et al., 2020). Samples were randomly chosen from both the baseline DDPM and our model, adjusting sampling timesteps across datasets to form synthetic sets. By examining the FID (Fréchet distance) measure as a timestep function on these datasets, we determined optimal sampling times. Our model consistently reached optimal FID scores in fewer timesteps than the U-Net-based model (Table 1), indicating faster convergence by our continuous U-Net-based approach.

To compute the FID, we generated two datasets, each containing 30,000 generated samples from each of the models, in the same way as we generated the images shown in the figures above. These new datasets are then

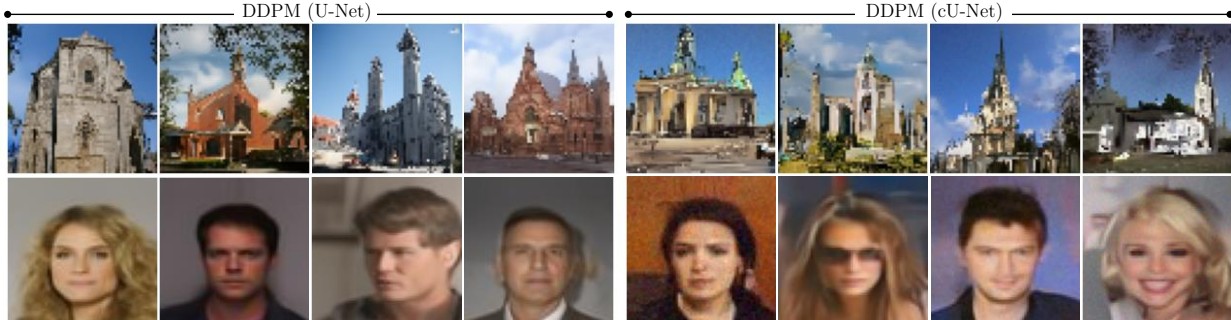

Figure 3: Randomly selected generated samples by our model (right) and the baseline U-Net-based DDPM (left) trained on CelebA and LSUN Church.

| Backbone | MNIST (28 × 28) | | | CelebA (64 × 64) | | | LSUN Church (128 × 128) | | |
|---|---|---|---|---|---|---|---|---|---|
| | FID ↓ | Steps ↓ | Time (s) ↓ | FID ↓ | Steps ↓ | Time (s) ↓ | FID ↓ | Steps ↓ | Time (s) ↓ |
| U-Net | 3.61 | 30 | 3.56 | 19.75 | 100 | 12.48 | 12.28 | 100 | 12.14 |
| U-Net[†] | 3.73 | 11 | 1.12 | 19.54 | 30 | 3.08 | 11.89 | 40 | 4.25 |
| cU-Net | 2.98 | 5 | 0.54 | 21.44 | 80 | 7.36 | 12.14 | 90 | 8.33 |
| cU-Net[†] | 2.55 | 2 | 0.18 | 21.41 | 15 | 1.37 | 11.77 | 30 | 2.68 |

Table 1: Performance metrics across datasets: FID scores (measured every 5 steps for CelebA and LSUN, and at every step for MNIST), sampling timesteps (Steps), and average generation time for both the U-Net and continuous U-Net (cU-Net) models. † indicates that the model used a DDIM sampler at inference time rather than the traditional DDPM sampler. As shown, the efficiency benefits are maintained across the different samplers.

directly used for the FID score computation with a batch size of 512 for the feature extraction. We also note that we use the 2048-dimensional layer of the Inception network for feature extraction as this is a common choice to capture higher-level features.

We examined the average inference time per sample across various datasets (Table 1). While both models register similar FID scores, our cU-Net infers notably quicker, being about 30% to 80% faster[2]. Notably, this enhanced speed and synthesis capability is achieved with marked parameter efficiency as discussed further in Section 5.3.

## 5.2 Image Denoising

Denoising is essential in diffusion models to approximate the reverse of the Markov chain formed by the forward process. Enhancing denoising improves the model's reverse process by better estimating the data's conditional distribution from corrupted samples. More accurate estimation means better reverse steps, more significant transformations at each step, and hence samples closer to the data. A better denoising system, therefore, can also speed up the reverse process and save computational effort.

In our experiments, the process of noising images is tied to the role of the denoising network during the reverse process. These networks use timesteps to approximate the expected noise level of an input image at a given time. This is done through the time embeddings which help assess noise magnitude for specific timesteps. Then, accurate noise levels are applied using the forward process to a certain timestep, with images gathering more noise over time. Figure 4 shows the process of noise accumulation in images over time, which is central to the functionality of diffusion models. By visualising noise at different timesteps, we demonstrate the correlation between timestep advancement and noise intensity. This not only validates the

---

[2]Note that inference times reported for both models were measured on a CPU, as current Python ODE-solver packages do not utilise GPU resources effectively, unlike the highly optimised code of conventional U-Net convolutional layers.

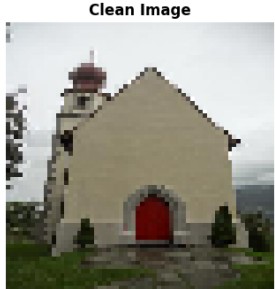 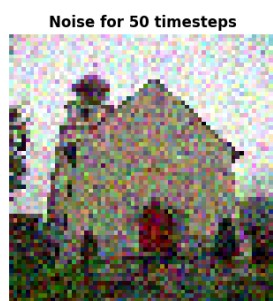 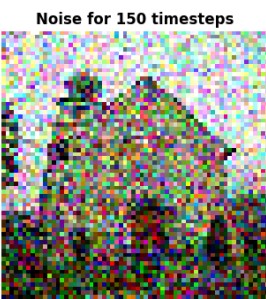 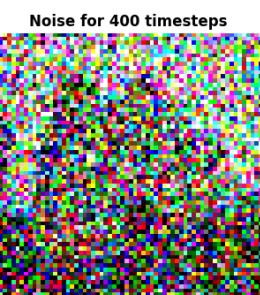

**Clean Image** — **Noise for 50 timesteps** — **Noise for 150 timesteps** — **Noise for 400 timesteps**

Figure 4: Visualisation of noise accumulation in images over increasing timesteps. As timesteps advance, the images exhibit higher levels of noise, showcasing the correlation between timesteps and noise intensity. The progression highlights the effectiveness of time embeddings in predicting noise magnitude at specific stages of the diffusion process.

| Noising Timesteps | Best SSIM Value | Best LPIPS Value |
|:---:|:---:|:---:|
| 50 | 0.88 / **0.90** | 0.025 / **0.019** |
| 100 | **0.85** / 0.83 | 0.044 / **0.038** |
| 150 | **0.79** / 0.78 | 0.063 / **0.050** |
| 200 | **0.74** / 0.71 | 0.079 / **0.069** |
| 250 | **0.72** / 0.64 | 0.104 / **0.084** |
| 400 | **0.58** / 0.44 | 0.184 / **0.146** |
| 600 | **0.44** / 0.26 | 0.316 / **0.238** |
| 800 | **0.32** / 0.18 | 0.419 / **0.315** |

Table 2: Comparative average denoising performance between U-Net (left values) and cU-Net (right values) for different noise levels over the test dataset. While U-Net predominantly achieves higher SSIM scores, cU-Net often outperforms LPIPS evaluations, indicating differences in the nature of their denoising approaches.

progression hypothesis of noise in the diffusion process but also the effectiveness of our continuous U-Net model's time embeddings.

In our denoising study, we evaluated 300 images for average model performance across noise levels, tracking SSIM and LPIPS over many timesteps to gauge distortion and perceptual output differences. Table 2 shows the models' varying strengths: conventional U-Net scores better in SSIM, while our models perform better in LPIPS. Despite SSIM being considered as a metric that measures perceived quality, it has been observed to have a strong correlation with simpler measures like PSNR (Horé & Ziou, 2010) due to being a distortion measure. Notably, PSNR tends to favour over-smoothed samples, which suggests that a high SSIM score may not always correspond to visually appealing results but rather to an over-smoothed image. This correlation underscores the importance of using diverse metrics like LPIPS to get a more comprehensive view of denoising performance.

The U-Net results underscore a prevalent issue in supervised denoising. Models trained on paired clean and noisy images via distance-based losses often yield overly smooth denoised outputs. This is because the underlying approach frames the denoising task as a deterministic mapping from a noisy image $y$ to its clean counterpart $x$. From a Bayesian viewpoint, when conditioned on $x$, $y$ follows a posterior distribution:

$$q(x|y) = \frac{q(y|x)q(x)}{q(y)}. \tag{15}$$

With the L2 loss, models essentially compute the posterior mean, $\mathbb{E}[x|y]$, elucidating the observed over-smoothing. As illustrated in Fig. 5 (and further results in Appendix A), our model delivers consistent detail preservation even amidst significant noise. In fact, at high noise levels where either model is capable of recovering fine-grained details, our model attempts to predict the features of the image instead of prioritising the smoothness of the texture like U-Net.

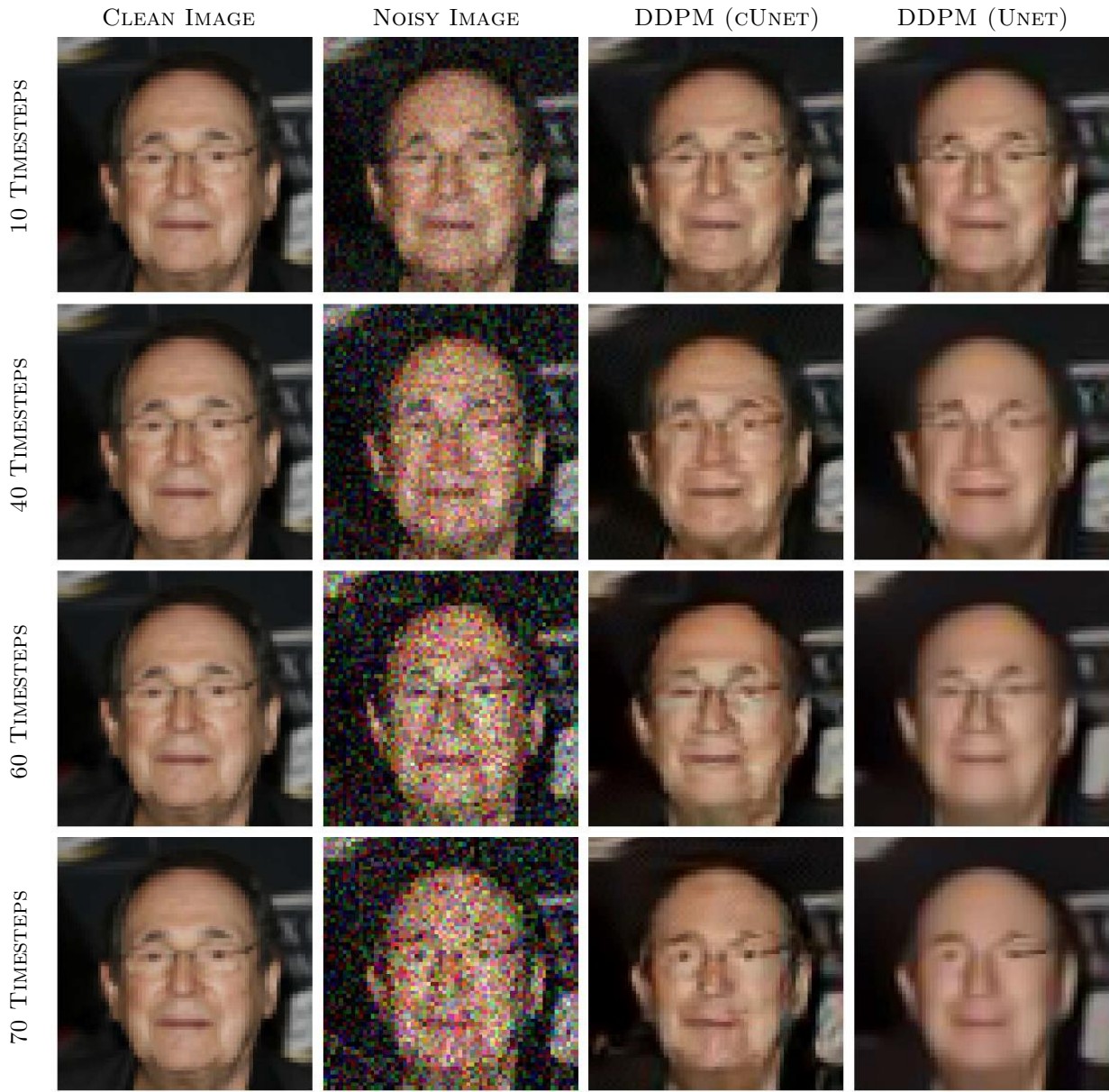

Figure 5: Original image (left), with Gaussian noise (second), and denoised using our continuous U-Net (third and fourth). As noise increases, U-Net struggles to recover the fine-grained details such as the glasses.

Furthermore, Figures 10 and 11 in Appendix B depict the *Perception-Distortion tradeoff*. Intuitively, this is that averaging and blurring reduce distortion but make images look unnatural. As established by (Blau & Michaeli, 2018), this trade-off is informed by the total variation (TV) distance:

$$d_{\text{TV}}(p_{\hat{X}}, p_X) = \frac{1}{2} \int |p_{\hat{X}}(x) - p_X(x)| \, dx, \tag{16}$$

where $p_{\hat{X}}$ is the distribution of the reconstructed images and $p_X$ is the distribution of the natural images. The perception-distortion function $P(D)$ is then introduced, representing the best perceptual quality for a given distortion $D$:

| Noise Steps | Best SSIM Step | Time SSIM (s) | Best LPIPS Step | Time LPIPS (s) |
|:---:|:---:|:---:|:---:|:---:|
| 50 | 47 / **39** | 5.45 / **4.40** | 41 / **39** | 4.71 / **4.40** |
| 100 | 93 / **73** | 19.72 / **9.89** | 78 / **72** | 16.54 / **9.69** |
| 150 | 140 / **103** | 29.69 / **14.27** | 119 / **102** | 25.18 / **13.88** |
| 200 | 186 / **130** | 39.51 / **18.16** | 161 / **128** | 34.09 / **17.82** |
| 250 | 232 / **154** | 49.14 / **21.59** | 203 / **152** | 43.15 / **21.22** |
| 400 | 368 / **217** | 77.33 / **29.60** | 332 / **212** | 69.77 / **29.19** |
| 600 | 548 / **265** | 114.90 / **35.75** | 507 / **263** | 106.42 / **35.49** |
| 800 | 731 / **284** | 153.38 / **39.11** | 668 / **284** | 140.26 / **39.05** |

Table 3: Comparison of average performance for U-Net (left) and cU-Net (right) at different noise levels in terms of the specific timestep at which peak performance was attained and time taken. These results are average across all the samples in our test set.

| METHOD | 50 Timesteps | | 150 Timesteps | | 400 Timesteps | |
|:---:|:---:|:---:|:---:|:---:|:---:|:---:|
| | SSIM | LPIPS | SSIM | LPIPS | SSIM | LPIPS |
| BM3D | 0.74 | 0.062 | 0.26 | 0.624 | 0.06 | 0.977 |
| Conv AE | 0.89 | 0.030 | 0.80 | 0.072 | 0.52 | 0.204 |
| DnCNN | 0.89 | 0.026 | **0.81** | 0.051 | 0.53 | 0.227 |
| Diff U-Net | 0.88 | 0.025 | 0.79 | 0.063 | **0.58** | 0.184 |
| Diff cU-Net | **0.90** | **0.019** | 0.78 | **0.050** | 0.44 | **0.146** |

Table 4: Comparative average performance of various denoising methods at select noise levels across the test set. Results demonstrate the capability of diffusion-based models (Diff U-Net and Diff cU-Net) in handling a broad spectrum of noise levels without retraining.

$$P(D) = \min_{p_{\hat{X}|Y}} d_{\text{TV}}(p_{\hat{X}}, p_X) \quad \text{s.t.} \quad \mathbb{E}[\Delta(X, \hat{X})] \leq D. \tag{17}$$

In this equation, the minimization spans over estimators $p_{\hat{X}|Y}$, and $\Delta(X, \hat{X})$ characterizes the distortion metric. Emphasizing the convex nature of $P(D)$, for two points $(D_1, P(D_1))$ and $(D_2, P(D_2))$, we have:

$$\lambda P(D_1) + (1 - \lambda)P(D_2) \geq P(\lambda D_1 + (1 - \lambda)D_2), \tag{18}$$

where $\lambda$ is a scalar weight that is used to take a convex combination of two operating points. This convexity underlines a rigorous trade-off at lower $D$ values. Diminishing the distortion beneath a specific threshold demands a significant compromise in perceptual quality. Additionally, the timestep at which each model achieved peak performance in terms of SSIM and LPIPS was monitored, along with the elapsed time required to reach this optimal point. Encouragingly, our proposed model consistently outperformed in this aspect, delivering superior inference speeds and requiring fewer timesteps to converge. These promising results are compiled and can be viewed in Table 3.

We benchmarked the denoising performance of our diffusion model's reverse process against established methods, including DnCNN (Zhang et al., 2017), a convolutional autoencoder, and BM3D (Dabov et al., 2007), as detailed in Table 4. Our model outperforms others at low timesteps in both SSIM and perceptual metrics. At high timesteps, while the standard DDPM with U-Net excels in SSIM, our cUNet leads in perceptual quality. Both U-Nets, pre-trained without specific noise-level training, effectively denoise across a broad noise spectrum, showcasing superior generalisation compared to other deep learning techniques. This illustrates the advantage of diffusion models' broad learned distributions for quality denoising across varied noise conditions.

## 5.3 Efficiency

Deep learning models often demand substantial computational resources due to their parameter-heavy nature. For instance, in the Stable Diffusion model (Rombach et al., 2022b) — a state-of-the-art text-to-image

diffusion model — the denoising U-Net consumes roughly 90% (860M of 983M) of the total parameters. This restricts training and deployment mainly to high-performance environments.

The idea of our framework is to address this issue by providing a plug-and-play solution to improve parameter efficiency significantly. Figure 6 illustrates that our cUNet requires only 8.8M parameters, roughly a quarter of a standard UNet. Maintaining architectural consistency across comparisons, our model achieves this with minimal performance trade-offs. In fact, it often matches or surpasses the U-Net in denoising capabilities.

While our focus is on DDPMs, cUNet's modularity should make it compatible with a wider range of diffusion models that also utilize U-Net-type architectures, making our approach potentially beneficial for both efficiency and performance across a broader range of diffusion models. CUNet's efficiency, reduced FLOPs, and memory conservation (Table 5) could potentially offer a transformative advantage as they minimize computational demands, enabling deployment on personal computers and budget-friendly cloud solutions.

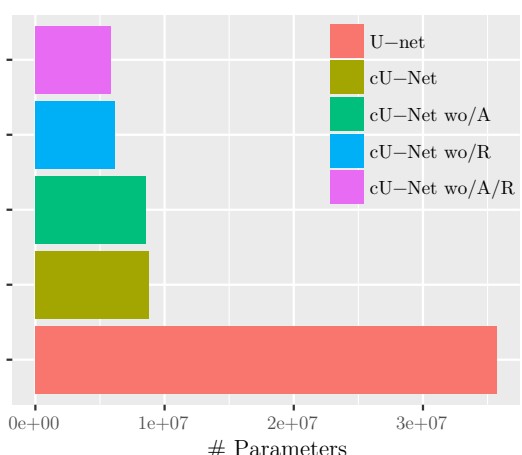

Figure 6: Total number of parameters for U-Net and continuous U-Net (cU-Net) models and variants. Notation follows Table 5.

| DDPM MODEL CONFIGURATION | GFLOPS | MB |
|---|---|---|
| U-Net | 7.21 | 545.5 |
| Continuous UNet (cU-Net) | **2.90** | **137.9** |
| cU-Net wo/A (no attention) | 2.81 | 128.7 |
| cU-Net wo/R (no resblocks) | 1.71 | 92.0 |
| cU-Net wo/A/R (no attention & no resblocks) | 1.62 | 88.4 |

Table 5: Number of GigaFLOPS (GFLOPS) and Megabytes in Memory (MB) for Different Models.

## 6 Conclusion

We explored the scalability of continuous U-Net architectures, introduction attention mechanisms, residual connections, and time embeddings tailored for diffusion timesteps. Through our ablation studies, we empirically demonstrated the benefits of the incorporation of these new components, in terms of denoising performance and image generation capabilities (Appendix C). We propose and prove the viability of a new framework for denoising diffusion probabilistic models in which we fundamentally replace the undisputed U-Net denoiser in the reverse process with our custom continuous U-Net alternative. In contrast with the exiting work, this new denoising network features a unique second-order Neural ODE block with residual connections and time embeddings, enhancing efficiency in our advanced diffusion models that utilize a deep implicit U-Net structure. As shown above, this modification is not only theoretically motivated, but is substantiated by empirical comparison. We compared the two frameworks on image synthesis, to analyse their expressivity and capacity to learn complex distributions, and denoising in order to get insights into what happens during the reverse process at inference and training. Our innovations offer notable efficiency advantages over traditional diffusion models, reducing computational demands and hinting at possible deployment on resource-limited devices due to their parameter efficiency while providing comparable synthesis performance and improved perceived denoising performance that is better aligned with human perception. Considerations for future work go around improving the ODE solver parallelisation, and incorporating sampling techniques to further boost efficiency.

## Acknowledgements

SCO gratefully acknowledges the financial support of the Oxford-Man Institute of Quantitative Finance. A significant portion of SCO's work was conducted at the University of Cambridge, where he also wishes to thank the University's HPC services for providing essential computational resources. CWC gratefully acknowledges funding from CCMI, University of Cambridge. GY was supported in part by the ERC IMI (101005122), the H2020 (952172), the MRC (MC/PC/21013), the Royal Society (IEC/NSFC/211235), the NVIDIA Academic Hardware Grant Program, the SABER project supported by Boehringer Ingelheim Ltd, Wellcome Leap Dynamic Resilience, and the UKRI Future Leaders Fellowship (MR/V023799/1). JH was supported in part by the Imperial College Bioengineering Department PhD Scholarship and the UKRI Future Leaders Fellowship (MR/V023799/1). CBS acknowledges support from the Philip Leverhulme Prize, the Royal Society Wolfson Fellowship, the EPSRC advanced career fellowship EP/V029428/1, EPSRC grants EP/S026045/1 and EP/T003553/1, EP/N014588/1, EP/T017961/1, the Wellcome Innovator Awards 215733/Z/19/Z and 221633/Z/20/Z, CCMI and the Alan Turing Institute. AAR gratefully acknowledges funding from the Cambridge Centre for Data-Driven Discovery and Accelerate Programme for Scientific Discovery, made possible by a donation from Schmidt Futures, ESPRC Digital Core Capability Award, and CMIH and CCIMI, University of Cambridge.

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

## A Appendix

This appendix serves as the space where we present more detailed visual results of the denoising process for both the baseline model and our proposal.

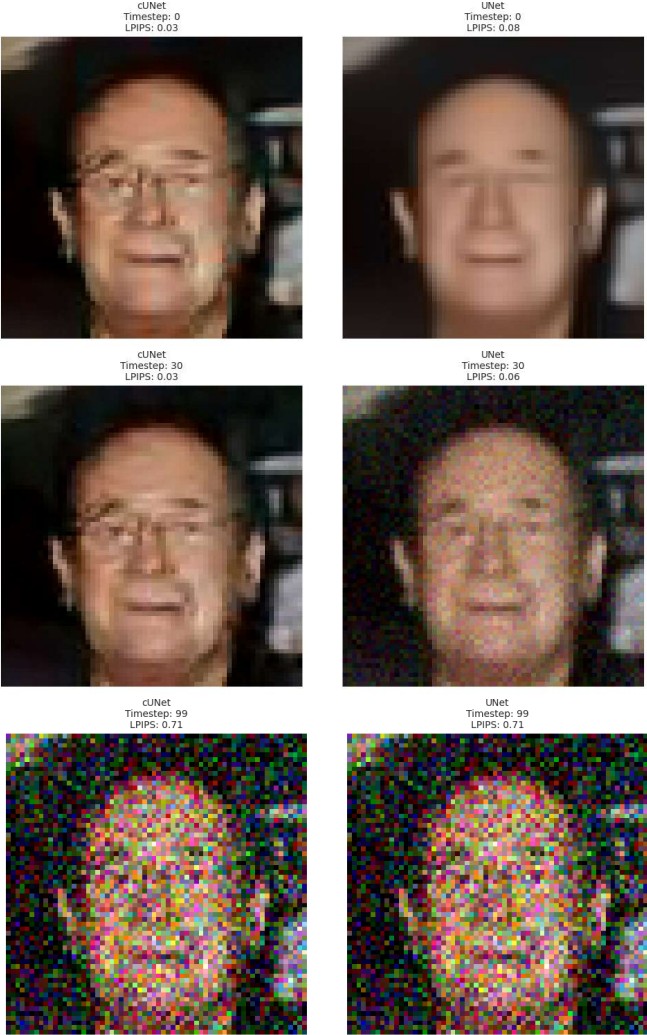

Figure 7: Tracking intermediate model denoising predictions. The images on the left are the outputs of our continuous U-Net, and the ones on the right are from the conventional U-Net.

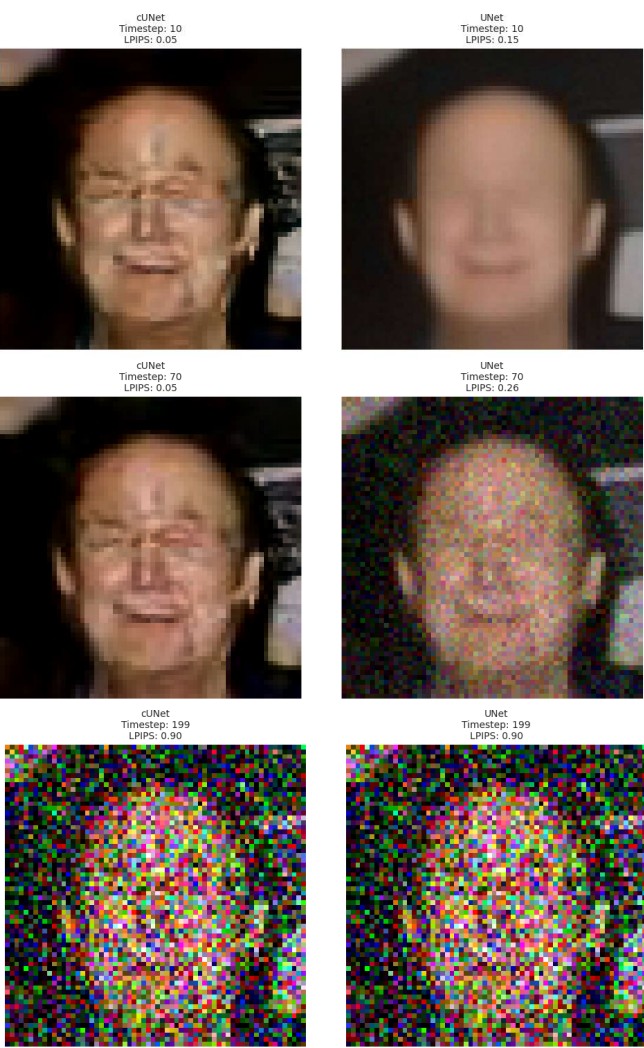

Figure 8: Tracking intermediate model denoising predictions: The images on the left depict the outputs of our continuous U-Net, which successfully removes noise in fewer steps compared to its counterpart (middle images) and maintains more fine-grained detail. The images on the right represent outputs from the conventional U-Net.

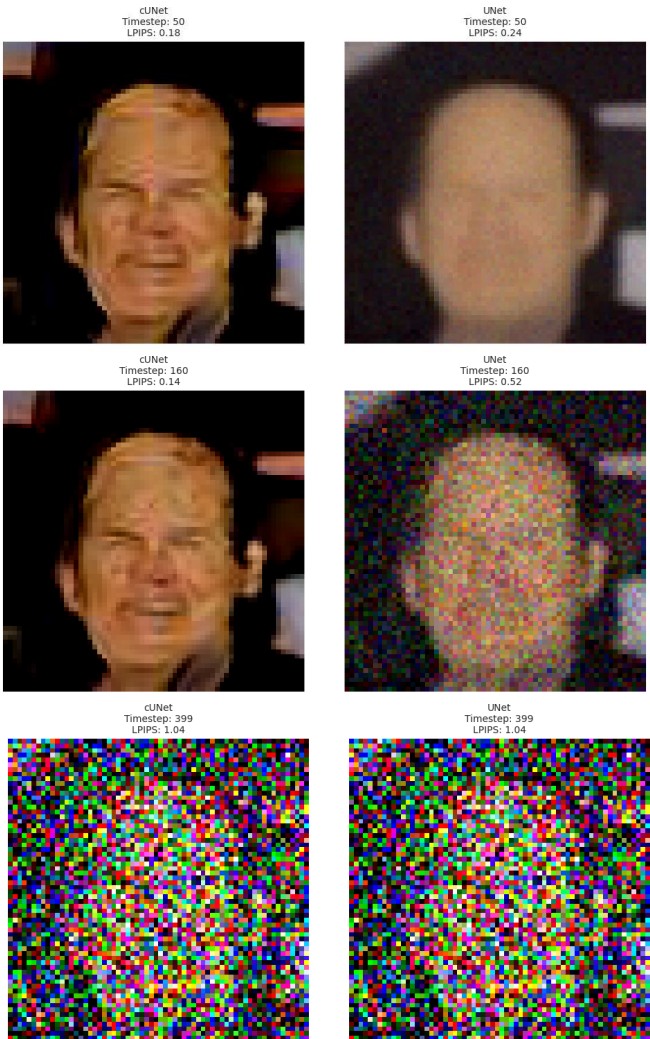

Figure 9: Tracking intermediate model denoising predictions: The images on the left depict the outputs of our continuous U-Net, which attempts to predict the facial features amidst the noise. In contrast, the images on the right, from the conventional U-Net, struggle to recognise the face, showcasing its limitations in detailed feature reconstruction.

## B  Appendix

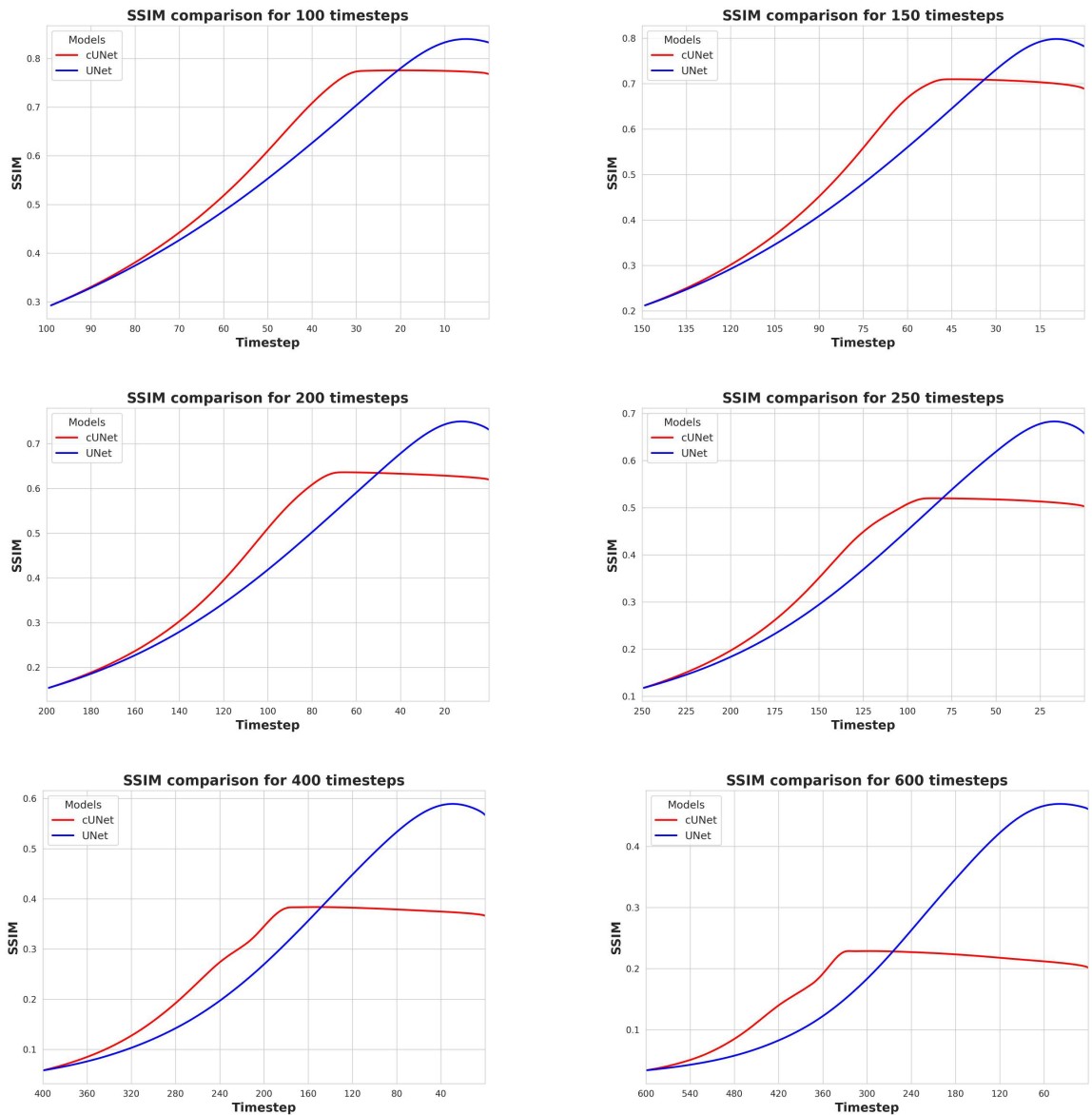

Figure 10: SSIM scores plotted against diffusion steps for varying noise levels for one image. The graph underscores the consistently superior performance of U-Net over cU-Net in terms of SSIM, particularly at high noise levels. This dominance in SSIM may be misleading due to the inherent Distortion-Perception tradeoff and the tendency of our model to predict features instead of distorting the images.

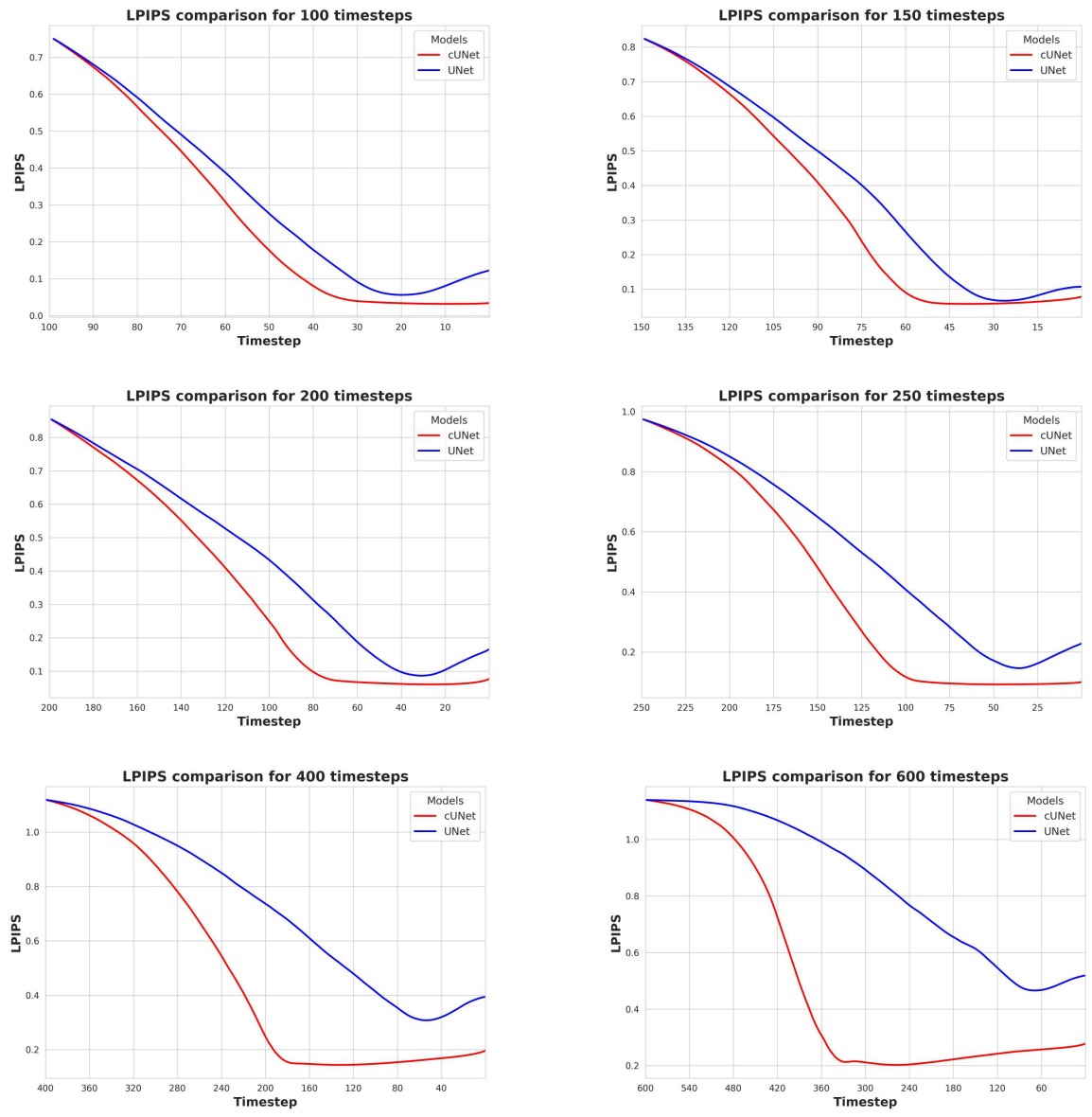

Figure 11: LPIPS score versus diffusion - or denoising timesteps for one image. We can observe how our continuous U-Net consistently achieves better LPIPS score and does not suffer from such a significant *elbow effect* observed in the U-Net model in which the quality of the predictions starts deteriorating after the model achieves peak performance. The discontinuous lines indicate the timestep at which the peak LPIPS was achieved. Here, we see that, especially for a large number of timesteps, continuous U-Net seems to converge to a better solution in fewer steps. This is because of the ability it has to predict facial features rather than simply settling for over-smoothed results (Figure 9).

## C  Appendix

In this short appendix, we showcase images generated for our ablation studies. As demonstrated below, the quality of the generated images is considerably diminished when we train our models without specific components (without attention and/or without residual connections). This leads to the conclusion that our enhancements to the foundational blocks in our denoising network are fundamental for optimal performance.

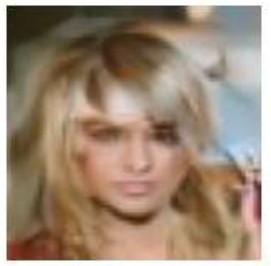 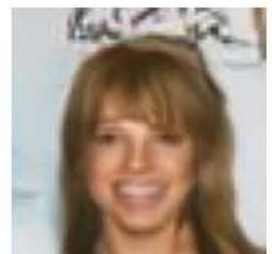 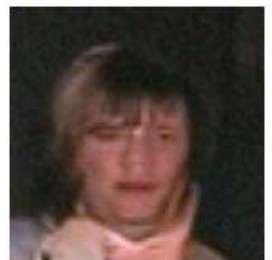 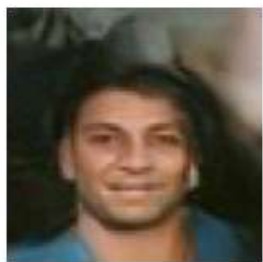

Figure 12: Representative samples from the version of our model trained without attention mechanism. The decrease in quality can often be appreciated in the images generated.

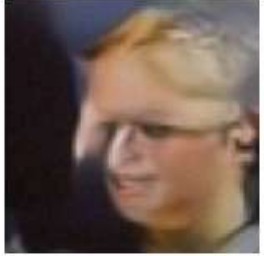 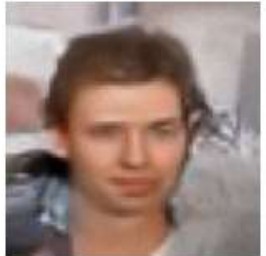 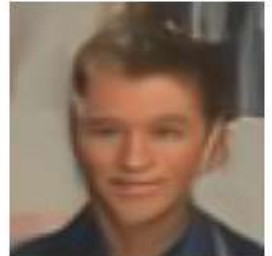 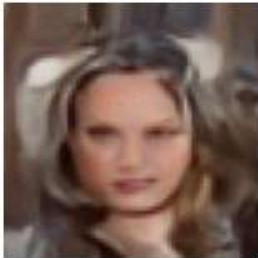

Figure 13: Representative samples from the version of our model trained without residual connections within our ODE block. We can see artefacts and inconsistencies frequently.

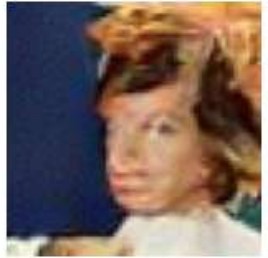 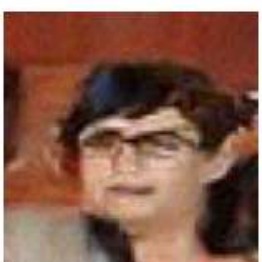 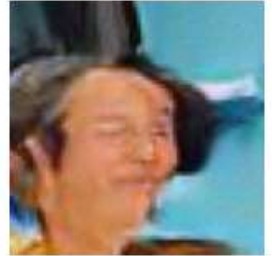 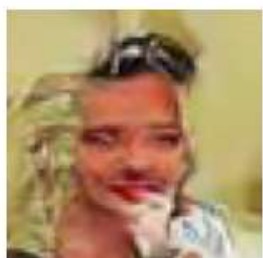

Figure 14: Representative samples from the basic version of our model which only includes time embeddings. As one can appreciate, sample quality suffers considerably.

