# OpenReview forum: "The Missing U for Efficient Diffusion Models"
_TMLR — Accepted by TMLR_

### Review · Reviewer_Duy9 · 2024-01-08

**Summary Of Contributions:**

This paper proposes to use continuous unets and the probability flow ODE to achieve faster and more parameter efficient sampling from diffusion models when compared with the DDPM baseline.

**Audience:**

Yes

**Claims And Evidence:**

No

**Requested Changes:**

See weaknesses

**Strengths And Weaknesses:**

Strengths
-----------
* There are some interesting questions about architecture raised in this manuscript, namely whether or not we can cut down the size of these existing diffusion models.

Weaknesses
---------------
* Overall, I found it very hard to parse what the exact contribution of the paper is. I believe that it's mostly the architecture, but the section on the probability flow ODE seems to deviate from this rather heavily.
* Why the continuous UNet? It seems like the continuous UNet just replaces the resnet structure of the internal blocks with a Neural ODE, but doing this should be very slow since one has to solve the forward + reverse differential equation for each block for training. Furthermore, Neural ODEs are constrained when compared with regular resnet blocks since they are diffeomorphic, and our UNet is supposed to model the score function, which is not diffeomorphic like a pure flow.
* Related to the above, it would help if the authors could rewrite the paper to talk exactly about what's modelled exactly. Equations (3) and (4) seem to be doing this, but it's not clear/a bit self-contradictory. In particular, (3) says that we're modeling the noise prediction, but (4) and the paragraph afterwards say that we are modeling the step between $x_{t + 1} and x_t$.
* The connection between the second-order solver and the probability flow ODE is not clear. The probability flow ODE (following Song "Score-Based Generative Modeling...") is a continuous dynamical system that arises after one learns the score (gradients) of the probability distribution through score matching (which is simulation-free). The paper here seems to be proposing a way to parameterize this ODE directly and to train it with a neural ODE training objective, but that has nothing to do with the probability flow ODE and is more generally a neural ODE objective.
* The experimental results are extremely out of date, as the baselines seem to be from the original DDPM paper. In particular, the diffusion literature has seen a lot of updating (including different architectures, a continuous training objective, fixing up core problems in the sampling process, etc...) so the paper should compare with these other components in place.
* There are a lot of irrelevant parts in the experimental section. For example, Figure 4 just shows a noising process, which is very well known and not relevant. Figure 5 just shows that the proposed image denoiser recovers fine details like glasses compared with the regular diffusion for an example, but this is somewhat arbitrary. In fact, the entire denoising section seems somewhat weird (especially with a lack of diffusion baselines like https://arxiv.org/abs/2201.11793). Generally, I would want the experimental section to be focused on the main claims of the paper (better image generation at a lower parameter cost), and these other tertiary experiments seem rather unnecessary given this.

---

> ### Author Response · Authors · 2024-02-19
> **Reply to Reviewer Duy9 (1/2)**
>
> We thank the reviewer for their thoughtful feedback. In our detailed response, we meticulously address each of the requested changes with our justifications, hoping to fully meet the reviewer's concerns and demands.
>
> > Overall, I found it very hard to parse what the exact contribution of the paper is. I believe that it's mostly the architecture, but the section on the probability flow ODE seems to deviate from this rather heavily.
>
> ➡️ We thank the reviewer for the comment. The paper's main contribution is indeed the introduction of a novel architecture for diffusion models that aims to make them faster and lighter. While the architecture is central, the discussion on the probability flow Ordinary Differential Equation (ODE) is essential to understanding the theoretical underpinning of how the proposed model achieves efficiency and speed. This section might seem distinct from the architecture discussion but it is integral in explaining the model's operational framework and its advantages over traditional approaches. To address the reviewer's concern, we have updated the manuscript adding clarity on the contribution and purpose of the probability flow. These changes can be seen at the end of Section 3.3 in blue colour.
>
> > Why the continuous UNet? It seems like the continuous UNet just replaces the resnet structure of the internal blocks with a Neural ODE, but doing this should be very slow since one has to solve the forward + reverse differential equation for each block for training...
>
> ➡️ Thanks for the comment. We would like to clarify that the implementation of a continuous UNet and the use of Neural ODEs within our architecture, it's essential to highlight that our motivation extends beyond merely replacing ResNet structures with Neural ODEs, which is far from being trivial. The decision to utilise Neural ODEs is strategically rooted in their capability to significantly reduce computational cost without increasing it. This reduction is achieved through the decreased necessity for storing active functions and leveraging the adjoint sensitivity method, which guarantees O(1) memory cost regardless of model complexity. This approach inherently builds reversibility into the architecture, ensuring efficient memory usage and substantially reducing computation time (theoretically), and is also demonstrated empirically in our results (see Table 1). The choice of Neural ODEs, therefore, is not just a theoretical preference but a practical one, aimed at addressing specific challenges associated with the computational and memory demands of traditional architectures while maintaining the model's ability to effectively model the score function. We have now added more clarity to the paper by adding a discussion at the beginning of section 3, which changes can be seen in blue colour.
>
> > Related to the above, it would help if the authors could rewrite the paper to talk exactly about what's modelled exactly. Equations (3) and (4) seem to be doing this, but it's not clear/a bit self-contradictory. In particular, (3) says that we're modeling the noise prediction, but (4) and the paragraph afterwards say that we are modeling the step between $x_{t+1}$ and $x_t$.
>
> ➡️ We thank the reviewer for the comment. Equation (3)  represents the continuous U-Net function, which models the transition probabilities using the dynamics of the system over time. Equation (4), on the other hand, defines the second-order derivative of the state with respect to time, which encapsulates the system's acceleration and is a function of both the current state and its velocity. The paragraph following Equation (4) aims to articulate how these equations model the step between noise predictions. We have now updated the paper to ensure that the distinction between noise prediction and transition steps is presented. The changes can be seen in blue colour.
>
> > The connection between the second-order solver and the probability flow ODE is not clear...
>
> ➡️ We thank the reviewer for their comment. We would like to clarify that our work aims to draw a conceptual link between the second-order differential equations used in our solver and the dynamics described by the probability flow ODE. While the probability flow ODE, as detailed by Song et al., is derived from the score of the data distribution using forward and reverse SDEs, our approach diverges by parameterising the dynamics via a second-order ODE that specifically models acceleration only in the reverse process. This approach, while inspired by the probability flow ODE, captures the complex dynamics itself in a different manner, offering a distinct perspective on modelling the evolution of the probability landscape. We have updated the paper to reflect this distinction– notably, the changes can be found in the 4th paragraph of Section 1 (see changes in blue).

---

> > ### Author Response · Authors · 2024-02-19
> > **Reply to Reviewer Duy9 (2/2)**
> >
> > > The experimental results are extremely out of date, as the baselines seem to be from the original DDPM paper. In particular, the diffusion literature has seen a lot of updating (including different architectures, a continuous training objective, fixing up core problems in the sampling process, etc...) so the paper should compare with these other components in place.
> >
> > ➡️ Thank you for the feedback. We appreciate the vast diffusion model research, including the adoption of a continuous training objective and enhancements to the sampling process. We would like to emphasise that such developments are complementary to our work rather than directly comparable. Our focus is on accelerating diffusion models through our continuous architecture. The chosen baselines, while appearing traditional, were selected for their relevance to our acceleration objective. We aim to complement, not replace, the existing methods that address different aspects of diffusion model performance. As such, our comparison with discrete U-Net versions is intended to highlight the specific advances our model contributes in terms of speed and efficiency. Other techniques/mechanisms with different purposes will lead to an unfair comparison. To clarify this point, we have updated the discussion of our goal and future work in the Related Work section of the paper (see changes in blue).
> >
> > > There are a lot of irrelevant parts in the experimental section. For example, Figure 4 just shows a noising process, which is very well known and not relevant. Figure 5 just shows that the proposed image denoiser recovers fine details like glasses compared with the regular diffusion for an example, but this is somewhat arbitrary. In fact, the entire denoising section seems somewhat weird (especially with a lack of diffusion baselines like https://arxiv.org/abs/2201.11793). Generally, I would want the experimental section to be focused on the main claims of the paper (better image generation at a lower parameter cost), and these other tertiary experiments seem rather unnecessary given this.
> >
> > ➡️ We are thankful for the feedback and the reference suggested by the reviewer.
> >
> > __[Baselines]__ We want to clarify the intention and scope of our work. Our objective is to introduce a paradigm shift in diffusion models by developing a new family of continuous networks. This stands in contrast to the discrete networks that are the same across all existing diffusion models. The referenced work, provided by the reviewer, focuses on the sampling strategy, which, although valuable, diverges from our work's primary aim of innovating the core architecture itself. The comparison between sampling strategies and architectural design is not parallel and thus is not directly applicable. The most appropriate baseline for our purpose remains the discrete version of U-Net, as it aligns with our goal of accelerating diffusion models through continuous networks.
> >
> > __[Figures]__ We appreciate your perspective on Figure 4. While the noising process is indeed well-established, this figure supports illustrating the effectiveness of our continuous network's time embeddings in predicting noise magnitude across different stages. However, we acknowledge that a better description of the purpose is needed. Therefore, we have updated the discussion on the figure in the manuscript– changes can be seen in blue colour. Thank you for your observation regarding Figure 5. The selection of the image denoising example, featuring the recovery of fine details such as glasses, was intentional and serves to illustrate the capabilities of our continuous U-Net architecture. This particular example was chosen to visually highlight the strength of our technique in preserving high-frequency details, which are often lost in diffusion processes. It is not an arbitrary choice, but rather a deliberate one to showcase the practical advantages of our model.
> >
> > We sincerely hope our detailed responses have satisfactorily addressed all the concerns and doubts raised by the reviewer. We are grateful for the opportunity to improve our manuscript based on your valuable feedback.

---

### Review · Reviewer_WHDr · 2024-01-20

**Summary Of Contributions:**

Summary: The authors introduce a continuous denoiser architecture for diffusion generative modeling by leveraging ideas from Neural ODEs. More specifically, the authors adapt the existing U-Net denoiser commonly used in diffusion modeling to a continuous U-Net based on second-order ODE dynamics. Figure 2 provides a nice intuition regarding the same. Empirical results demonstrate the effectiveness of the continuous denoiser.

**Audience:**

Yes

**Claims And Evidence:**

Yes

**Requested Changes:**

Requested Changes:

1. In section 3.3, the authors mention: “Our architecture outperformed DDPMs in terms of efficiency and accuracy”. For readability, It would be nice to provide a reference to a table or figure which justifies this argument. Moreover, the theoretical argument presented regarding faster but less accurate solutions using the probability flow ODE seems a bit redundant considering the analysis presented in [2]. Lastly, I had some trouble understanding the relevance of the derivation presented in Proposition 3.1 in the context of the proposed methodology. It would be nice if the authors could intuitively elaborate upon the same in their response and subsequently in the revision.

2. Missing citations and discussion of existing work: Since there is a large body of work on speeding up diffusion models, I found the discussion in the introduction a bit dated as the discussion of several important directions in fast sampling in diffusion models is missing. For instance,
   1. Discussion of training-free samplers like DPM-Solver [3], DEIS [4], EDM [2] is missing.
   2. Discussion of methods which accelerate diffusion models by incorporating conditional information like [5, 6] or perform diffusion in the latent space [7, 8].
   3. Discussion of more efficient ways for diffusion: EDM [2], CLD [9], PSLD[10].
   4. The authors discuss ES-DDPM. In addition I think its worth mentioning works like Truncated DDPM [11] and Denoising Diffusion GANs [12] which are based on similar ideas of truncating the forward process of DDPMs for efficient sampling.

It is worth noting that these works are an example and the authors should look into the literature for more relevant works. Moreover, I think the paper currently lacks a dedicated related works section and discussion of these works in more detail in this section would be ideal.

3. The claim in the introduction: “However, models trained on continuous timesteps often underperform compared to discretely-trained models” seems incorrect. See Section 4.4 and Table 3 in Song et al where models trained with continuous time objective outperform models trained with a discrete time objective. I see that the authors cite the Song et al. paper so perhaps they meant continuous time models outperform discrete ones?
Furthermore, the authors mention “training must be repeated for each desired step count”. This needs further elaboration since for discrete or continuous time diffusion, as long as the number of noise levels sampled during training are fairly large (around N=1000 to 4000), we can get a decent score model. This allows sampling using a lot less noise levels during inference than used during training. Therefore, this point needs to be clarified in more detail in the paper.

Minor Comments:
The line “In general, these methods provide inference-time…” is repeated twice in the main text.

References
[1] Continuous U-Net: Faster, Greater and Noiseless [Cheng et al.]

[2] Elucidating the Design Space of Diffusion-Based Generative Models [Karras et al.]

[3] DPM-Solver: A Fast ODE Solver for Diffusion Probabilistic Model Sampling in Around 10 Steps [Lu et al.]

[4] Fast Sampling of Diffusion Models with Exponential Integrator [Zhang et al.]

[5] Diffusion Autoencoders: Toward a Meaningful and Decodable Representation
[Preechakul et al.]

[6] DiffuseVAE: Efficient, Controllable, and High-Fidelity Generation from Low-Dimensional Latents [Pandey et al.]

[7] Score-based Generative Modeling in Latent Space [Vahdat et al.]

[8] High-Resolution Image Synthesis with Latent Diffusion Models [Rombach et al.]

[9] Score-Based Generative Modeling with Critically-Damped Langevin Diffusion [Dockhorn et al.]

[10] A Complete Recipe for Diffusion Generative Models [Pandey et al.]

[11] Truncated Diffusion Probabilistic Models and Diffusion-based Adversarial Auto-Encoders [Zheng at al.]

[12] Tackling the Generative Learning Trilemma with Denoising Diffusion GANs [Xiao et al.]

**Strengths And Weaknesses:**

Strengths:
1. Given, the widespread adoption of diffusion models for generative modeling, I think the main strength of the paper is the reduction in wall clock time (Table 1), and computational and memory footprint (Table 5) for generating samples as compared to the standard UNet architecture.
2. Since the authors propose updates in the denoiser, I expect the proposed approach to be complementary to existing techniques for fast diffusion sampling and it would be interesting to benchmark the continuous denoiser with fast sampling methods like DDIM, DPM-Solver, etc.

Weaknesses:

1. The paper has limited methodological novelty in the sense that the authors largely reuse methodological ideas presented in another paper [1]. However, based on TMLR’s acceptance criterion, this is not a problem per se.

2. See Requested Changes for more details

---

> ### Author Response · Authors · 2024-02-19
> **Reply to Reviewer WHDr**
>
> We thank the reviewer for their insightful feedback. In our response, we will address each requested change one by one, providing our justifications with the hope of addressing the concerns and demands effectively.
>
> > __Requested Change 1:__
>
> ➡️ Firstly, we thank the reviewer for the comment and we have added a reference to Table 1 at the end of the “Our architecture outperformed DDPMs in terms of efficiency and accuracy” sentence for better readability. Secondly, the goal of our theoretical formulation is that we would like to show that the probability flow of ODE is faster than SDE. Although it is shown in [2], this is an essential assumption of our analysis. Once we know this fact, then it’s helpful for our later analysis. In the later part of the analysis, we focus on second-order neural ODEs in the diffusion model performing faster than first-order neural ODEs. We understand that the explanation of Proposition 3.1 is not enough to understand how’s the relationship of this paper. We added an extra explanation and how it’s connected to the probability ODE flow. Firstly, the primary purpose of proposition 3.1 is going to show the probability flow can be transformed into first-order ODE. Why do we want to transform it into first-order ODE? The reason is that we want to use the adjoint method for backpropagation. The use of the adjoint method can reduce the memory cost and force it into constant memory cost due to the trace operation. We updated the paper for a clearer explanation and indicated blue colours.
>
> > __Requested Change 2:__
>
> ➡️ We are thankful for the reviewer's suggestion. We added all of the suggested references and more in the introduction and the related work section (all changes are in blue).
>
> > __Requested Change 3:__
>
> ➡️ We thank the reviewer for pointing out this typo. Yes, the continuous time models outperform discrete ones. We have changed the word from underperformed to outperformed and removed the  “training must be repeated for each desired step count” to avoid any confusion.
>
> > __Minor comments:__
>
> We have fixed the typo of the repeated sentence. We thank the reviewer for spotting this issue.
>
> We hope our responses have adequately addressed all concerns and doubts raised by the reviewer. We deeply appreciate the valuable feedback provided and look forward to further guidance.

---

### Review · Reviewer_DR5X · 2024-02-05

**Summary Of Contributions:**

This paper introduces a more efficient architecture for diffusion-based generative model, especially for denoising diffusion models. This approach featuring a continuous U-Net denoising network, namly c-U-Net, significantly reduces the number of parameters and computational costs, achieving faster convergence and improved noise robustness. The authors demonstrate its effectiveness through experiments and corresponding analysis, showing that their model outperforms standard U-Nets in both efficiency and quality of generated images on various datasets under the DDPM setting.

**Audience:**

Yes

**Broader Impact Concerns:**

No significant broader impact concerns were found in the paper.

**Claims And Evidence:**

No

**Requested Changes:**

1. The differences and connections compared with Cheng et al., 2023 and Karras et al., 2022 need to be clarified.

2. The presentation regarding the introduction and method sections needs to be improved.

3. The experiments, including experimental setting, comparison with recent models and comparison across different resolutions are suggested.

Please see my detailed comments in weakness part.

**Strengths And Weaknesses:**

### **Strength**
- The proposed method is overall simple but effective. It does not require additional tuning of hyper-parameters and make the model with lighter weight. Moreover, the computation cost is reduced and the denoising is more efficient as we can see the model can output less blurry predictions of $x_0$ at high noisy-level.

- The experimental results well validate the approach. Notably, the paper also provides comprehensive study to across different sampling timesteps, with different metrics, etc.

### **Weakness**
I have several concerns regarding the novelty, clarity of the method, design of the experiments and some defaults in the presentation. Please see the points below:

- It seems this work is heavily built upon Cheng et al., 2023. It is not clear how this work differs from Cheng et al., 2023 and tackles the problem in diffusion models.

- The SDEs can be generalized as a sum of PF ODE and a Langevin diffusion SDE is not novel, as it is already shown in the EDM paper (Karras et al., 2022). The authors may need to clarify the difference between the analysis in section 3.3 those in EDM.

- It is unclear how the Unet architecture is modified from Figure 2. How does the ODE block shown in the right panel play a role in the entire network and how does the left panel illustrate the design?

- The notation of x" in Eq (4) is not consistent the one in the text below Eq (4). $X_0$,  $g(\cdot, \cdot)$ and $\theta_g$ are not explained.

- Can the authors specify the resolution of the images used for the training? As a study of the network architecture, it is also important the proposed design can be efficient across different dimensionality (image resolution).

- In the sampling stage, the authors seem to only compare when adopting DDPM sampler. It would be interesting if the authors could show some comparison using different samplers to demonstrate the efficiency of the proposed c-U-Net.

- One possible reason of the improvement in computation efficiency can be the reduction of parameter size. (Please note this is not necessarily a weakness.) If the authors would like to highlight the computation efficiency is improved, it would be better to compare GFLOPs with Unet has similar parameter size.

- Some recent works have also been working on tackling the efficiency of the diffusion model architecture. How does the proposed model work compared with them? (Wang et al, 2022; Wang et al., 2023; Zheng et al., 2023a; Gao et al., 2023; Ding et al., 2023; Arakawa et al., 2023; Zheng et al., 2023b)

- Some claims in the paper are misleading. For example, the authors point out "However, models trained directly on continuous  timesteps often underperform compared to discretely-trained models (Song et al., 2020b)" in the introduction. I did not find related evidence in the ScoreSDE paper (Song et al., 2020b). On one hand, recent empirical results have shown models on continuous time reach state-of-the-art performance in Karras et al, 2022. Moreover, in Song et al., 2020b, the connection between score sde and ddpm has been derivated. It would be better if the authors could provide additional evidence to support the claim here or clarify in order to avoid potential misunderstanding.

- The review of existing literatures are not comprehensive. For example,
  - Several works have proposed to truncate the diffusion chain and start the generation from an implicit distribution instead of standard Gaussian noise  (Xiao et al., 2021; Pandey et al., 2022; Zheng et al., 2023c; Lyu et al., 2022).
  - Besides Bao et al., 2022, a lot of works have been working on employing advanced numerical solvers for SDE/ODE (Liu et al., 2022; Lu et al., 2022; Zhang & Chen, 2023; Karras et al., 2022).
  - Using manifold constraints or inverse problem hypothesis for diffusion models also involves several representative papers (Lou et al., 2020; Kawar et al., 2022; Chunget al., 2022; Giannis, et al., 2023; Rout et al., 2023; Lou and Ermon, 2023)

### References

- Wang, W., Bao, J., Zhou, W., Chen, D., Chen, D., Yuan, L., & Li, H. (2022). Sindiffusion: Learning a diffusion model from a single natural image. arXiv preprint arXiv:2211.12445.

- Wang, Z., Jiang, Y., Zheng, H., Wang, P., He, P., Wang, Z., ... & Zhou, M. (2023). Patch diffusion: Faster and more data-efficient training of diffusion models. arXiv preprint arXiv:2304.12526.

- Zheng, H., Nie, W., Vahdat, A., Azizzadenesheli, K., & Anandkumar, A. Fast sampling of diffusion models via operator learning. In International Conference on Machine Learning (pp. 42390-42402). PMLR, 2023a.

- Gao, S., Zhou, P., Cheng, M. M., & Yan, S. (2023). Masked diffusion transformer is a strong image synthesizer. arXiv preprint arXiv:2303.14389.

- Ding, X., Wang, Y., Xu, Z., Welch, W. J., & Wang, Z. J. (2022). Continuous conditional generative adversarial networks: Novel empirical losses and label input mechanisms. IEEE Transactions on Pattern Analysis and Machine Intelligence.

- Arakawa, S., Tsunashima, H., Horita, D., Tanaka, K., & Morishima, S. (2023). Memory Efficient Diffusion Probabilistic Models via Patch-based Generation. arXiv preprint arXiv:2304.07087.

- Zheng, H., Wang, Z., Yuan, J., Ning, G., He, P., You, Q., Yang, H., and Zhou, M. Learning stackable and skippable LEGO bricks for efficient, reconfigurable, and variable-resolution diffusion modeling, 2023b

- Song, Y., Sohl-Dickstein, J., Kingma, D. P., Kumar, A., Ermon, S., & Poole, B. Score-based generative modeling through stochastic differential equations. arXiv preprint arXiv:2011.13456, 2020b

- Xiao, Z., Kreis, K., & Vahdat, A. (2021). Tackling the generative learning trilemma with denoising diffusion gans. arXiv preprint arXiv:2112.07804.

- Pandey, K., Mukherjee, A., Rai, P., and Kumar, A. DiffuseVAE: Efficient, controllable and high-fidelity generation from low-dimensional latents. arXiv preprint arXiv:2201.00308, 2022.

- Zheng, H., He, P., Chen, W., and Zhou, M. Truncated diffusion probabilistic models and diffusion-based adversarial auto-encoders. In The Eleventh International Conference on Learning Representations, 2023c

- Liu, L., Ren, Y., Lin, Z., and Zhao, Z. Pseudo numerical methods for diffusion models on manifolds. In International Conference on Learning Representations, 2022.

- Lu, C., Zhou, Y., Bao, F., Chen, J., Li, C., and Zhu, J. DPM-solver: A fast ODE solver for diffusion probabilistic model sampling in around 10 steps. In Oh, A. H., Agarwal, A., Belgrave, D., and Cho, K. (eds.), Advances in Neural Information Processing Systems, 2022.

- Zhang, Q. and Chen, Y. Fast sampling of diffusion models with exponential integrator. In The Eleventh International Conference on Learning Representations, 2023

- Karras, T., Aittala, M., Aila, T., and Laine, S. Elucidatingthe design space of diffusion-based generative models. In Proc. NeurIPS, 2022

- Lou, A., Lim, D., Katsman, I., Huang, L., Jiang, Q., Lim, S. N., & De Sa, C. M. (2020). Neural manifold ordinary differential equations. Advances in Neural Information Processing Systems, 33, 17548-17558.

- Kawar, B., Elad, M., Ermon, S., & Song, J. (2022). Denoising diffusion restoration models. Advances in Neural Information Processing Systems, 35, 23593-23606.

- Chung, H., Kim, J., Mccann, M. T., Klasky, M. L., & Ye, J. C. (2022). Diffusion posterior sampling for general noisy inverse problems. arXiv preprint arXiv:2209.14687.

- Daras, G., Shah, K., Dagan, Y., Gollakota, A., Dimakis, A. G., & Klivans, A. (2023). Ambient Diffusion: Learning Clean Distributions from Corrupted Data. arXiv preprint arXiv:2305.19256.

- Rout, L., Raoof, N., Daras, G., Caramanis, C., Dimakis, A. G., & Shakkottai, S. (2023). Solving linear inverse problems provably via posterior sampling with latent diffusion models. arXiv preprint arXiv:2307.00619.

- Lou, A., & Ermon, S. (2023). Reflected diffusion models. arXiv preprint arXiv:2304.04740.

----------

*Given the aforementioned concerns, my current inclination is to rate this submission below the acceptance threshold. However, I am open to reconsidering my evaluation if the authors address the highlighted issues or provide clarifications in their rebuttal.*

---

> ### Author Response · Authors · 2024-02-18
> **Reply to Reviewer DR5X (1/3)**
>
> We greatly appreciate the time and effort the reviewer has dedicated to providing their insightful feedback and constructive comments on our manuscript. In this reply, we intent to address each and every point highlighted in the "Weaknesses" section of the review above.
>
> > It seems this work is heavily built upon Cheng et al., 2023. It is not clear how this work differs from Cheng et al., 2023 and tackles the problem in diffusion models.
>
> ➡️ We thank the reviewer for the comment. We now clarify that our work distinctly advances the state of the art by integrating Temporal Dynamics within the continuous U-Net architecture, a feature not present in the work of Cheng et al., 2023. Our integration of time embeddings allows our model to finely tune its response to the temporal aspects of the diffusion process, an enhancement that is critical for denoising in DDPMs. Furthermore, we have introduced sophisticated attention mechanisms and residual connections into the continuous U-Net framework. These features are designed to capture complex long-range dependencies, which are essential for reconstructing images with a high degree of detail and coherence. Such architectural innovations are specifically tailored to address and overcome the limitations of current diffusion models, marking a significant departure from the foundations laid by Cheng et al. and making our approach uniquely equipped to handle the intricacies of diffusion-based generative modelling. We now summarise three key distinctions.
>
> i) We redesigned the architecture to suit denoising tasks, involving changes in output channels, loss functions, and stride adjustments for minimal spatial resolution loss. ii) We integrated time embeddings to enable the model to accurately model and adapt to the diffusion process over time, a critical aspect for denoising in DDPMs. iii) We integrated time embeddings to enable the model to accurately model and adapt to the diffusion process over time, a critical aspect for denoising in DDPMs. All these changes are far from being trivial to address.
>
> These innovations not only delineate our work from Cheng et al. but also introduce a first-of-its-kind architecture within the domain of diffusion models, demonstrating that cUNets are both scalable and efficient empirically. To address this question, we have updated the manuscript, and all changes can be seen in blue colour (beginning of Section 4 "Methodology").
>
> > The SDEs can be generalized as a sum of PF ODE and a Langevin diffusion SDE is not novel, as it is already shown in the EDM paper (Karras et al., 2022). The authors may need to clarify the difference between the analysis in section 3.3 those in EDM.
>
> ➡️ We thank the reviewer for their comment. We would like to clarify that our work aims to draw a conceptual link between the second-order differential equations used in our solver and the dynamics described by the probability flow ODE. While the probability flow ODE, as detailed by several papers including Song et al. and Karras et al., is derived from the score of the data distribution using forward and reverse SDEs, our approach diverges by parameterising the dynamics via a second-order ODE that specifically models acceleration only in the reverse process. This approach, while inspired by the probability flow ODE, captures the complex dynamics itself in a different manner.  Proposition 3.1 shows that the probability flow ODEs can be written as a first-order ODE which is novel in our paper. The analysis of this part explains why our second-order neural ODEs perform better in terms of speed. We have updated the paper to reflect this distinction– notably, the changes can be found in the 3rd paragraph of Section 1 and towards the end of section 4.3. (see changes in blue).
>
> > It is unclear how the Unet architecture is modified from Figure 2. How does the ODE block shown in the right panel play a role in the entire network and how does the left panel illustrate the design?
>
> ➡️ Thank you for your comment. We would like to clarify the architecture presented in Figures 1 and 2. Figure 1 provides an overarching view of the network, showcasing how the dynamic blocks function within the entire denoising framework of the DDPM. Specifically, these dynamic blocks are where the actual continuous transformation of the data occurs. On the other hand, Figure 2 details the internal workings of these blocks. The left panel of Figure 2 represents the ODE block within the continuous U-Net, illustrating the continuous nature of the transformations applied. The right panel delves deeper into the ODE function approximator, which is a crucial component of the ODE block and is responsible for the network's ability to adapt its processing at any given time, guided by the time embeddings. However, we recognise that the caption of Figure 2 would benefit of a better description. To address this question, we have updated the description and can be seen in blue.

---

> > ### Author Response · Authors · 2024-02-18
> > **Reply to Reviewer DR5X (2/3)**
> >
> > > The notation of $x''$ in Eq (4) is not consistent the one in the text below Eq (4), $X_0, g(., .)$ and $\theta_g$ and are not explained.
> >
> > ➡️ Thank you for highlighting the need for clearer notation. We have revised the manuscript to ensure consistency and clarity. Specifically, $x''$ denotes the second-order derivative representing acceleration in our second-order neural ODEs. The function $g$, with parameters $\theta_g$, approximates the initial velocity. The initial condition $X_0$ corresponds to the initial state $x_{\tilde{t}_0}$. Our model requires two initial conditions to capture the dynamics of the system accurately: the initial state and the initial velocity. These changes will help the readers in understanding the diffusion process as modeled by our neural ODE framework.
> >
> > > Can the authors specify the resolution of the images used for the training? As a study of the network architecture, it is also important the proposed design can be efficient across different dimensionality (image resolution).
> >
> > ➡️ We value the reviewer’s emphasis on the importance of image resolution for network architecture studies. To address this, our experiments included images of varying resolutions to ensure the robustness and efficiency of our proposed design across different dimensionalities. Specifically:
> >
> > 1. Low-Resolution Datasets: Initial validations were performed using the MNIST and Fashion-MNIST datasets, which consist of images of 28x28 pixels, to verify the model's efficacy in lower resolutions. Although for simplicity reasons results are shown in Table 1 generation samples were not included in the final version of the paper.
> > 2. Medium-Resolution Datasets: Further, we assessed the model on the CelebA dataset with images resized to 64x64 pixels.
> > 3. Higher-Resolution Datasets: The model was also tested on the LSUN Church dataset, with images resized to 128x128 pixels​​.
> >
> > We have explicitly mentioned the resolution of images for each dataset used in our experiments in the revised manuscript (headings of Table 1).
> >
> > > In the sampling stage, the authors seem to only compare when adopting DDPM sampler. It would be interesting if the authors could show some comparison using different samplers to demonstrate the efficiency of the proposed c-U-Net.
> >
> > ➡️ We appreciate the reviewer’s suggestion to examine the efficiency of the proposed continuous U-Net (cU-Net) across different sampling methods. Following this valuable input, we have conducted additional experiments employing the Denoising Diffusion Implicit Models (DDIM) sampler. These experiments were designed to further validate the robustness and efficiency of our cU-Net architecture.
> >
> > We have updated Table 1 in our manuscript with the new results derived from these experiments. The revised table now includes performance metrics using both the DDPM and DDIM samplers across the datasets we considered. Our findings show that cU-Net maintains its efficiency advantages across different samplers. In particular, the results demonstrate that our model consistently achieves faster convergence and maintains or improves upon the fidelity of generated samples when compared with the baseline U-Net, regardless of the sampler employed​​.
> >
> > > One possible reason of the improvement in computation efficiency can be the reduction of parameter size. (Please note this is not necessarily a weakness.) If the authors would like to highlight the computation efficiency is improved, it would be better to compare GFLOPs with Unet has similar parameter size.
> >
> > ➡️ We appreciate the reviewer's point regarding the comparison of computational efficiency in relation to parameter size. We understand the concern that comparing our cU-Net with a conventional U-Net of similar parameter size might seem like a direct route to demonstrate computational efficiency. However, such a comparison would overlook the primary advantage of our model, which is "parameter efficiency".
> >
> > Our model is designed to operate with fewer parameters while maintaining architectural equivalence with traditional U-Nets—having the same number of upsampling and downsampling blocks, attention layers, etc.—and achieving comparable or improved performance. This parameter efficiency is a key innovation of our work, as it allows for significant reductions in model size and computational demand without sacrificing output quality or architectural complexity. This is especially relevant when considering deployment in resource-constrained environments where model size and efficiency are of paramount importance. Crucially, if two models perform the same number of operations (or have the same number of parameters), the number of operations per floating point will indeed be similar. Therefore, while we acknowledge the reviewer's point that comparing GFLOPs against a similarly sized U-Net could be informative, such a comparison would not highlight the distinct advantages offered by the cU-Net’s framework.

---

> ### Author Response · Authors · 2024-02-18
> **Reply to Reviewer DR5X (3/3)**
>
> > Some recent works have also been working on tackling the efficiency of the diffusion model architecture. How does the proposed model work compared with them? (Wang et al, 2022; Wang et al., 2023; Zheng et al., 2023a; Gao et al., 2023; Ding et al., 2023; Arakawa et al., 2023; Zheng et al., 2023b)
>
> ➡️ We acknowledge the importance of situating our work within the context of recent efforts to enhance the efficiency of diffusion model architectures. As per the reviewer's suggestion, we have added a Related Work section before the conclusion of our manuscript. This section carefully considers the contributions of Wang et al. (2022, 2023), Zheng et al. (2023a, 2023b), Gao et al. (2023), Ding et al. (2023), and Arakawa et al. (2023), particularly focusing on how our continuous U-Net (cUNet) architecture intersects with, diverges, and complements these recent advancements justifying that the main point is not to compare techniques as they are orthogonal solutions.
>
> It is imperative to note that while the aforementioned works provide valuable improvements within the domain of diffusion models, they primarily build upon and propose enhancements to the foundational U-Net structure. Our approach, in contrast, introduces a fundamental change in the architecture of the denoiser. We utilise "continuous dynamical blocks," an innovation that significantly deviates from the standard U-Net design, resulting in improved parameter efficiency and computational performance without compromising denoising effectiveness​​.
>
> Our newly included Related Work section details how cUNet's architecture and its underpinning principles can potentially be integrated with the techniques proposed in the papers mentioned by the reviewer. This integration suggests that cUNet could complement and extend the efficiency gains achieved by these methods. We also clarify that our model's efficiency—characterized by reduced floating-point operations (FLOPs) and memory conservation—supports the adoption of cUNet across various platforms, including those with limited computational resources.
>
> > Some claims in the paper are misleading. For example, the authors point out "However, models trained directly on continuous timesteps often underperform compared to discretely-trained models (Song et al., 2020b)" in the introduction. I did not find related evidence in the ScoreSDE paper (Song et al., 2020b). On one hand, recent empirical results have shown models on continuous time reach state-of-the-art performance in Karras et al, 2022. Moreover, in Song et al., 2020b, the connection between score sde and ddpm has been derivated. It would be better if the authors could provide additional evidence to support the claim here or clarify in order to avoid potential misunderstanding.
>
> ➡️ We appreciate the reviewer's attention to the nuances of performance comparisons between continuous and discretely trained models in diffusion processes. Indeed models trained directly on continuous timesteps do not __underperform__ but __outperform__ compared to discretely-trained models as showcased by Song et al. and more recently by Karras et al. We amended this typo and to avoid any potential confusion, we have rephrased our statement in the manuscript (changes in blue in the 3rd paragraph, Section 1).
>
> > The review of existing literatures are not comprehensive...
>
> ➡️ We appreciate the reviewer’s extensive suggestion of missing literature and we modified our introduction (Section 1 - changes in blue) and added the Related Work section (Section 3) to include all of the references suggested in a coherent and informed manner.
>
> We hope that our changes and clarifications address all of the reviewer's doubts and concerns. We would like to, once more, thank reviewer D5RX for their kind comments and useful feedback

---

### Decision · Action_Editor_1yRC · 2024-03-26

**Recommendation:** Accept with minor revision

**Comment:**

All reviewers agree that the novelty of the paper is somewhat unclear. Following the recommendation of one of the reviewers, please follow the following suggestion:
"The clarification provided does highlight some differences, but I recommend that the authors further emphasize the innovative aspects of their work or its applicability to broader or different contexts than those covered by existing research. This could help in underlining the unique contributions of their study more clearly."

Please update the paper accordingly.

**Audience:**

The paper is definitely of interest to the TMLR audience.

**Claims And Evidence:**

The paper proposes a new architecture for DDGMs that is based on a novel denoising network. The claim of the paper is the following:
A new implicit denoising network based on a dynamic Neural ODE block is parameter-efficient, computationally efficient, converges faster, and allows for achieving good generative performance in image synthesis.

For all points, the authors presented empirical evidence. There are two main issues raised by the reviewers, regarding this claim:
- The comparison to other methods is limited.
- The novelty of the paper seems limited.

Since TMLR is a platform for sharing ideas rather than chasing SOTA, I do not find the first point problematic. However, indeed the novelty of the paper requires a better explanation.